# WE HAVE IT COVERED: A RESAMPLING-BASED METHOD FOR UPLIFT MODEL COMPARISON

## ABSTRACT

Uplift models play a critical role in modern marketing applications to help understand the incremental benefits of interventions and identify optimal targeting strategies. A variety of techniques exist for building uplift models, and it is essential to understand the model differences in the context of intended applications. The uplift curve is a widely adopted tool for assessing uplift model performance on the selection universe when observations are available for the entire population. However, when it is uneconomical or infeasible to select the entire population or a large simple random sample, it becomes difficult or even impossible to estimate the uplift curve without appropriate sampling design. To the best of our knowledge, no prior work has addressed how to construct uplift curve estimates on the entire population along with uncertainty quantifications, using only a population subsample. We propose a two-step sampling procedure and a resampling-based approach to compare uplift models with uncertainty quantification, examine the proposed method empirically through simulations and real data applications, and conclude with a discussion.

## 1 INTRODUCTION

Uplift modeling becomes increasingly popular in marketing and other domains in recent years (Baier & Stöcker, 2022; Gubela et al., 2017; Hu, 2023; Li & Zhu, 2024; Rzepakowski & Jaroszewicz, 2012). It involves a target population also referred to as the selection universe (e.g., all adults in the United States of America). Intervention(s) in consideration will be experimented on either a segment of the population or the entire population, and the uplift model aims to estimate the individual-level causal impact of the intervention(s) on some target outcome(s). Another research stream focusing on heterogeneous treatment effect (HTE) also targets the treatment difference on the individual level. Systematic review and benchmarking study on uplift modeling and HTE methods are available in (Rößler & Schoder, 2022; Zhang et al., 2021).

When we limit our scope to consider cases with a binary treatment variable and an uni-variate outcome variable, there are three main categories of methods to build an uplift model in the literature (Gutierrez & Gérardy, 2017; Zhang et al., 2021): 1) T-learners (Künzel et al., 2019) (also called "Two-model" approaches) which build response models for the treatment and control group separately first, and take the prediction difference from these two models as the predicted uplift, as well as the S-learners (Künzel et al., 2019) which includes the treatment indicator in the model. These methods don't directly optimize for uplift in the training process. 2) Class transformation approaches (Athey & Imbens, 2015; Jaskowski & Jaroszewicz, 2012; Li & Zhu, 2024), which utilize a transformation of the origination binary outcome variable to a new binary outcome variable, and translate the uplift estimation problem to the direct estimation of the conditional expectation of the transformed outcome variable given covariates; 3) Approaches (Guelman et al., 2015; Zhao et al., 2017) which aim to optimize for uplift directly in the training process. They are often modifications of existing machine learning methods with customized objective functions during the model training process.

When there are multiple candidate uplift models during the iterative model development process, and it is essential to determine the trajectory of model improvements based on quantitative evidence. As we cannot observe the responses of an individual under both treatment and control assignments, some commonly used metrics in machine learning such as mean squared error, log loss, area under

the curve (AUC) may not give us a good picture to the conversion uplift we are interested in the most. The uplift curve is a widely adopted tool for uplift model evaluation and comparison in the literature (Gutierrez & Gérardy, 2017; He et al., 2024; Rößler & Schoder, 2022; Verbeken et al., 2025), with the associated area under uplift curve (AUUC) metric.

To construct the uplift curve estimate with its definition, we need to have the responses for all individuals in the selection universe. However, in practice it is unlikely to include the entire population into a marketing campaign for efficiency or compliance concerns. For instance, in direct mail marketing, the cost is approximately linear to the number of mails being sent. When the eligible selection universe is vast, it can be very costly to obtain a large simple random sample (SRS) from the universe, not to mention to select the entire universe. Instead, one often ranks the individuals by the uplift metric first, and only select either a fixed number of candidates or all above a threshold for the marketing campaign. As uplift models produce different predictions and rankings, it is critical to know which model will lead to the highest gain in uplift for a given selection size.

In addition, there is a trade-off between exploration and exploitation in an iterative model development process that can leads to a biased/non-SRS sample from the population. One may start with a SRS from the universe and come up with several candidate uplift models from this sample, and select a few most promising ones. Then in order to maximize the total expected uplift given a fixed budget, it is optimal to use the best candidate models to perform ranking and select a fixed number of corresponding top candidates. This however would lead to a biased training dataset for the future, and we may need to include an additional representative subsample from the population. The combined subsample could still be a non-SRS from the population.

To the best of our knowledge, no prior work has addressed how to construct point estimates of the uplift curve on the entire population using a population subsample with uncertainty quantification to compare multiple uplift models. Our research aims to fill this gap and provide a solution to evaluate and compare the performance of uplift models in practice with uncertainty quantification.

The rest of the paper is organized as follows. We start with the problem formulation with proper notations. Next we propose a two-step sampling scheme and introduce the nested bootstrap estimation procedure to construct both point and interval estimates for the uplift curve(s). Lastly, we evaluate the performance of the proposed method via simulation studies, followed by two real data application examples and discussion.

## 2 PROBLEM FORMULATION

In this section, we introduce some general notations associated with uplift modeling and the problems under investigation.

Consider a population $\mathbb{P}$ with in total $N$ individuals, where $p$ attributes are observed for each individual. The attribute matrix is denoted as $X$ with dimension $N \times p$. We have a binary treatment variable $T$, where $T_i = 0$ indicates individual $i$ is assigned to the control group and $T_i = 1$ indicates $i$ is in the treated group, $i = 1, 2, ..., N$. We want to measure the causal impact of the treatment to an outcome variable $Y$ of interest, and in practice $Y$ could be a binary event such as making a purchase, registering an account or a continuous variable such as total transaction amount.

We could then express the uplift of the treatment to individuals with characteristics $X = x$ in Equation (1).

$$U(x) = E(Y|X = x, do(T = 1)) - E(Y|X = x, do(T = 0)) \tag{1}$$

When treatment allocation and X are independent, i.e., $T \perp\!\!\!\perp X$, we have

$$U(x) = E(Y|X = x, T = 1) - E(Y|X = x, T = 0) \tag{2}$$

We only consider the binary treatment option $T = 0$ or $1$ throughout this article and assume the treatment assignment $T$ is independent of $X$.

To express the uplift curve, we need to introduce a few more notations. Let $M$ denote a predictive model that assigns to each individual $i$ a score $\text{Score}_{M,i}$. The corresponding rank of individual $i$ under model $M$ is denoted by $\text{Rank}_{M,i}$. For a given cutoff $k \in \mathbb{N}$, define

$$A_{M,k} = \{ i : \text{Rank}_{M,i} \leq k \},$$

that is, $A_{M,k}$ represents the set of the top-$k$ individuals according to model $M$. We can define the theoretical cumulative gain of selecting set $A_{M,k}$ on $Y$ as $F_M(k)$:

$$F_M(k) = \sum_{i \in A_{M,k}} E[Y_i | do(T=1)] - \sum_{i \in A_{M,k}} E[Y_i | do(T=0)] \qquad (3)$$

As we are not able to assign both treatment and control to the same individual, given an randomized treatment assignment plan on the population, we could estimate $F_M(k)$ by

$$f_M(k) = \left( \frac{\sum_{T_i=1} Y_{i,k}}{N_k^{T=1}} - \frac{\sum_{T_i=0} Y_{i,k}}{N_k^{T=0}} \right)(N_k^{T=1} + N_k^{T=0}) = \left( \frac{\sum_{T_i=1} Y_{i,k}}{N_k^{T=1}} - \frac{\sum_{T_i=0} Y_{i,k}}{N_k^{T=0}} \right)k \quad (4)$$

where $N_k^{T=t}$ denotes the total number of candidates assigned to treatment option $T = t$, and $\sum_{T_i=t} Y_{i,k}$ denotes the sum of observed outcomes in each treatment group, $t = 0, 1$. The mean uplift in this case is defined as $(\sum_{T_i=1} Y_{i,k}/N_k^{T=1} - \sum_{T_i=0} Y_{i,k}/N_k^{T=0})$, which is essentially the mean observed difference of the treated group and control group within the top $k$ candidates. The cumulative gain equals to the mean uplift multiplied by selection size $k$.

To construct the uplift curve, the x-axis often consists of a series of selection sizes corresponding to the 0th, 5th, 10th, ..., 100th selection percentiles of the population, then on the y-axis the corresponding cumulative gain values will be plotted.

The Qini curve introduced in Radcliffe (2007) shares the same x-axis as the uplift curve, and it differs from the cumulative gain measure by a constant for a given selection size. The area under the uplift curve (AUUC) and the area under the Qini curve (Qini coefficient) are widely adopted uplift model evaluation metrics. When the treatment groups are randomly allocated, the Qini curve will be mostly proportional to the uplift curve as $N_k^{T=1}/k$ shall be nearly a constant across different selection sizes $k$ in practice. Thus we can focus on the uplift curve reconstruction in our discussion as we assume random treatment allocation.

Assume we have $S$ different uplift models $M_1, M_2, ..., M_S$ in consideration. When we select all $N$ candidates in the population, there will be no difference in the underlying cumulative gain $F_{M_s}(N)$ on $Y$ by using predictions from uplift model $M_s$ for ranking, $s = 1, ..., S$.

When we only select a portion of the universe, choosing which model to rank the candidates becomes a relevant question to optimize for total uplift. In order to get a cumulative gain estimate $f_{M_s}(n)$, technically we need to use model $M_s$ to rank the universe and select the top $n$ candidates. For multiple models, we can split the selection universe randomly to sub-universes of size $N_s$, where $N = \sum_{s=1}^{S} N_s$. Then we select the top $nN_s/N$ candidates ranked by model scores of $M_s$ in the sub-universe $s$ respectively to keep the same selection ratio $n/N$. The point estimates of the theoretical cumulative gain at selection percentile $n/N$ could then be obtained by plugging the observations into the equation (4). The sampling scheme is displayed in Figure 1, and this is typically adopted in practice to compare the mean uplift metrics of several models at a specified selection percentile.

Meanwhile, it seems difficult to draw statistical inference here. Let's consider the scenario with 2 candidate models $M_1$ and $M_2$ first. When sample size $n$ is large, one may try to construct an asymptotic confidence interval for $F_{M_1}(n) - F_{M_2}(n)$ by dividing the selected population based on ranking model and treatment assignment into 4 blocks: $(M_1, M_2) \times (T=0, T=1)$. Within each block $[M_s, T = t]$, $s = 1, 2$ and $t = 0, 1$, the individuals are assumed to be i.i.d from certain distribution with mean $\mu_{s,t}$, where $\mu_{s,t}$ is the mean of target variable $Y$ in the universe selected by model $M_s$ with treatment $T = t$. In this case $F_{M_1}(n) - F_{M_2}(n) = n[(\mu_{1,1} - \mu_{1,0}) - (\mu_{2,1} - \mu_{2,0})]$. Central limit theorem may be used to obtain the asymptotic distribution of the sample mean of each block, and an asymptotic confidence interval for the difference of the difference $(\mu_{1,1} - \mu_{1,0}) - (\mu_{2,1} - \mu_{2,0})$ may be constructed. However, this i.i.d. assumption may not hold, which invalidates the asymptotic confidence interval obtained above.

In addition, in order to compare the cumulative gain performance of these 2 ranking models at different choices of the selection size, we need to estimate the uplift curves on the entire selection universe. It seems difficult or nearly impossible to come up with proper mean uplift estimates at selection sizes greater than $n$ needed for the uplift curve construction, when only selecting $n$ top-ranked candidates from the universe. We need additional representative samples from the universe to help understand the performance at different spectrums as well to construct the uplift curves on the entire universe.

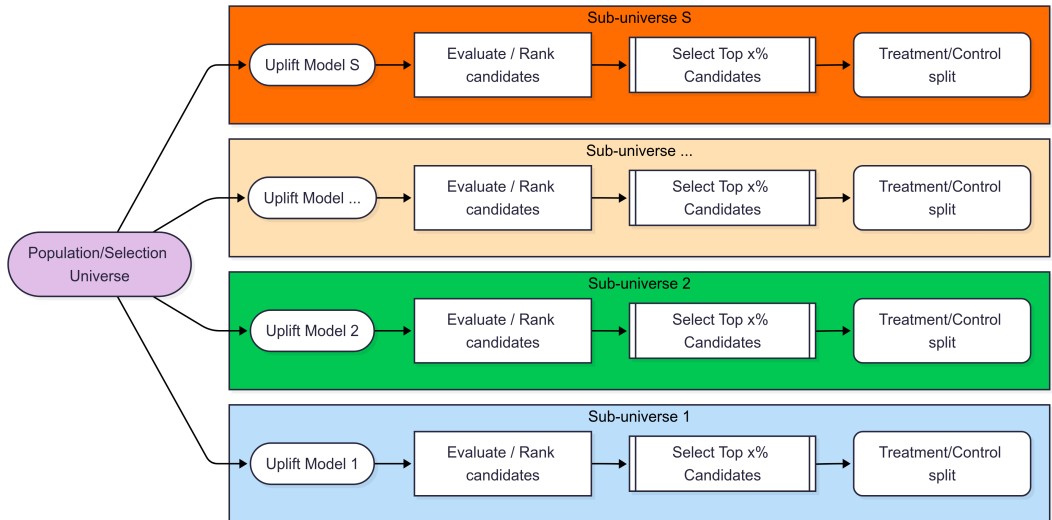

Figure 1: Randomized sample selection process flow solely driven by uplift model ranks.

## 3 METHOD

### 3.1 A TWO-STEP SAMPLING SCHEME

Instead of selecting only the top ranked candidates by the models in consideration, we propose a modified selection scheme to select $n$ samples via a simple random sampling step followed by a model rank based sampling procedure. Assume we want to evaluate $S$ candidate models $\{M_1, M_2, ..., M_S\}$, we can use a subset of models out of $S$ candidate models for rank based sampling. Without loss of generality, assume models $\{M_1, M_2, ..., M_{S_0}\}$ are used for rank-based sampling as we could relabel the models included for ranking otherwise. The sampling scheme is as follows:

- Step 1: Select a simple random sample with size $n_r$ from the selection universe with size N. Here $n_r > 0$;
- Step 2: Split the remaining universe randomly into $S_0$ sub-universes with sizes $N'_1, N'_2, ..., N'_{S_0}$, and select the top $(n - n_r)N'_s/(N - n_r)$ candidates ranked by model $s$ respectively, where $s = 1, ..., S_0$ and $\sum_{s=1}^{S_0} N'_s = N - n_r$;
- Step 3: Combined the samples in Steps 1 and 2 to construct the chosen set $C$.

The sampling procedure is displayed in Figure 2. Here we do not require $S_0 = S$, i.e. including all the $S$ candidate models in the rank-based sampling (Step 2), while we could still use the collected sample $C$ to construct uplift curve estimates for all $S$ candidate models using the methodology to be introduced in Section 3.2.

Note that the total selected sample size $|C| = n$. When $n_r$ is close to $n$, the average model rank in $C$ will be close to $N/2$, and the cumulative gain is likely much smaller than choosing $n$ samples with top model ranks if the model is well calibrated, which could be economically unsustainable to test. When $n_r$ is much less than $n$, we are still able to keep the vast majority of the $n$ top samples while introducing a small collection of simple random sample into $C$.

By including the simple random sampling step, the set $C$ constructed here is now a random sample from the population, with unequal inclusion probabilities when $n_r < n$. The inclusion probabilities are equal to 1 for subjects with top model ranks that are less than the selection size in each corresponding sub-universe, and for people with "next-tier" model ranks that are possible to be selected in Step 2 but not guaranteed, the inclusion probabilities are within $(n_r/N, 1)$. The minimum inclusion probability equals $n_r/N$ for people with low model ranks who could only be possibly selected in Step 1.

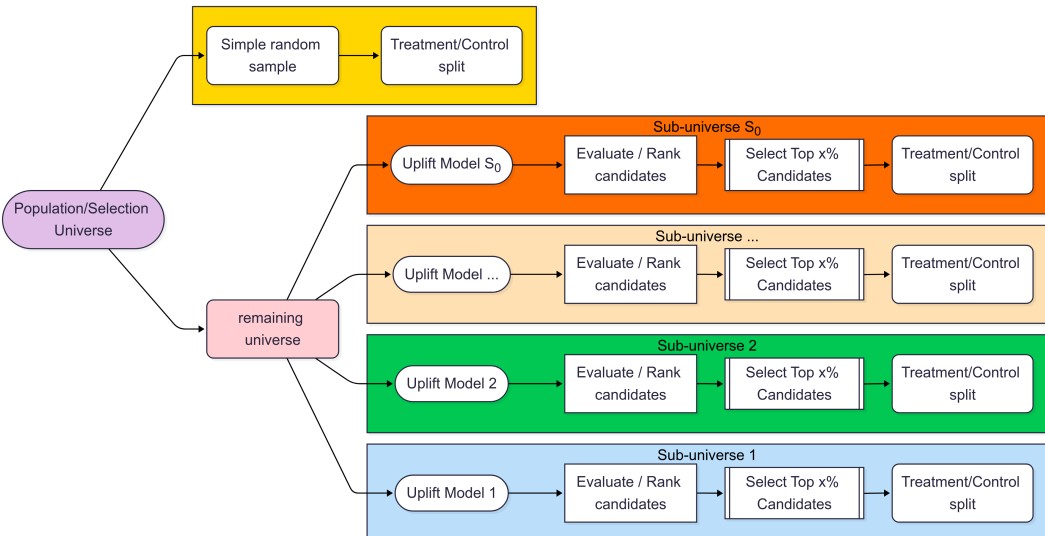

Figure 2: Proposed two-step sample selection scheme with randomization with $S_0$ ranking models.

The inclusion probability for each individual under the sampling scheme above could be expressed as a function of the model rank information from all $S_0$ models being considered for ranking, along with the selection size parameters $n_r, n, N, N'_1, ..., N'_{S_0}$. When $S_0 = 1$, it can be shown that for an individual $i$ with model rank $m_i$, the inclusion probability

$$P(i \in C)$$
$$= \begin{cases} n_r/N, & \text{if } m_i > n; \\ 1, & \text{if } m_i <= n - n_r; \\ n_r/N + (1 - n_r/N) \sum_{j=m_i-(n-n_r)}^{\min(m_i-1, n_r)} \binom{m_i-1}{j}\binom{N-m_i}{n_r-j}/\binom{N-1}{n_r} & \text{if } n \geq m_i \geq n - n_r + 1. \end{cases}$$

The inclusion probability formula is more complicated for the cases with $S_0 \geq 2$ ranking models. The derivation is included in the Appendix Section A.1.

## 3.2 NESTED BOOTSTRAP ESTIMATION

As we consider a randomized treatment allocation scheme, in practice it could be conducted as follows: we take a sample $C$ using the aforementioned two-step scheme from the universe $\mathbb{P}$, and we assign treatment options $T$ randomly on the entire universe $\mathbb{P}$, each individual will only receive 1 treatment option $T_i = 1$ or $0$. We use $\mathcal{W}_i = (X_i, T_i, Y_i)$ to represent each individual $i$'s full information: characteristics, the treatment received and the corresponding outcome. We can only observe the full information on selected set $C$, while for the others we know $X$ and the planned assignment $T$.

For each model $s = 1, ..., S$, we could use its model score to rank the individuals in $\mathbb{P}$ and then use formula (4) to express the cumulative gain at each selection size as a function of $\{\mathcal{W}_i\}_{i=1}^N$. These point estimates obtained via $\{\mathcal{W}_i\}_{i=1}^N$ represent the values derived from the definitions when full information of the entire universe $\mathbb{P}$ is available.

Now let us try to restore the uplift curves for all $S$ models on $\mathbb{P}$ using the selected sample $C$ only. Denote the inclusion probability for individual $i$ as $p_i$. The uplift curve consists of an array of (selection size, cumulative gain) pairs up to the size of the entire selection universe. One may think of constructing an inverse probability weighted (IPW) point estimator used in sampling (Lohr, 2021) for the cumulative gain at each selection size using the inverse inclusion probability as weights, however its form is rather complicated when the selected sample $C$ is not a simple random sample of the population, and it is even harder to derive proper interval estimates theoretically. The naive bootstrap method for uncertainty quantification is also not applicable here as it requires a SRS from the population.

To overcome this difficulty, here we propose the following nested bootstrap procedure in Algorithm 1 to construct both point and interval estimates of the uplift curve for all $S$ models.

---

**Algorithm 1:** Nested Bootstrap Procedure

**Input:** The selected sample $C$ from the two-step sampling scheme above and the inclusion probabilities $p_i$ for each $i \in C$.

1 **for** *each outer bootstrap iteration $b = 1, ..., B$,* **do**

2      Take a sample of size $n$ from $C$ with replacement and equal sample weights. Denote the sample set as $C_b$;

3      **for** *each inner bootstrap iteration $d = 1, ..., D$,* **do**

4          Take a sample of size $N$ on set $C_b$ with replacement, using inverse inclusion probabilities $1/p_i$ as weights. Denote the sample set as $H_{b,d}$; **for** *each model $s, s = 1, ..., S$,* **do**

5              Treat $H_{b,d}$ as a "pseudo" bootstrap sample of the population full information $\{\mathcal{W}_i\}_{i=1}^N$. Perform model ranking and use formula (4) to get the cumulative gain estimates at different selection percentiles to construct the uplift curve estimate $\mathcal{C}_{b,d}^s$;

6      Consolidate uplift curve estimates $\mathcal{C}_{b,d}^s, d = 1, ..., D$ by taking the median value of the cumulative gain estimates at each selection size to get the aggregated uplift curve estimate $\mathcal{C}_b^s$ for each $s = 1, ..., S$;

**Output:** For each $s = 1, ..., S$, we have $B$ uplift curve bootstrap estimates $\{\mathcal{C}_b^s\}_{b=1}^B$.

---

At each selection percentile of the uplift curve, we could take the 2.5% and 97.5% empirical percentiles of the corresponding cumulative gain from $B$ aggregated uplift curve estimates $\{\mathcal{C}_b^s\}_{b=1}^B$ to construct a point-wise 95% confidence band of the uplift curve, and take the median value as the final point estimate of the uplift curve for model $s, s = 1, ..., S$.

Here are some intuitions behind the nested bootstrap procedure. For the outer bootstrap step, we sample with replacement with equal sample weights on the selected set $C$ and keep the same sample size $n$ as $C$, such that the bootstrap samples $\{C_b\}_{b=1}^B$ will properly estimate the sampling distribution's variability of a given statistic constructed with $C$. The inner bootstrap essentially tries to construct statistics to estimate the mean uplift at each selection percentile to restore the uplift curve. It is done by firstly performing an inverse inclusion probability reweighted over-sampling on the bootstrap set $C_b$, with sample size equal to the population size $N$ and with replacement. Then plug-in estimates are obtained with these "pseudo" bootstrap samples, and the median is chosen as the statistic to stabilize the estimate. The reweighted resampling with inverse inclusion probability as weights accounts for the sampling design and make the pseudo bootstrap sample approximate the population better than equal weight bootstrap.

## 4 SIMULATION STUDY

In this section, we will illustrate the performance of the proposed method with simulated data to compare 2 uplift models, while the comparison could be performed in a pair-wise fashion when there are more than 2 models potentially with some multiplicity control. In the two-step sampling scheme, only the ranks from model 1 will be used to get model rank based samples in Step 2, i.e. $S_0 = 1$.

### 4.1 SIMULATION SETUP

The true underlying data generating process is assumed as follows. Consider a binary outcome variable $Y_i$, where $P(Y_i = 1) = 1/\{1 + \exp[-f(\boldsymbol{X_i}, \epsilon_i, T_i)]\}$ for some function $f$, and $\boldsymbol{X_i} = (X_1, ..., X_Q)$ represents the observable characteristics, $\epsilon_i \sim N(0, \sigma^2)$ denotes the unobservable random factors in the data generating process, $T_i = 0$ or 1 denotes the treatment assignment.

We consider $Q = 40$ variables $X_1, ..., X_{40}$ drawn from a multivariate normal distribution with mean vector $\mu = \mathbf{0}$, and covariance matrix $\Sigma = 0.2\mathbf{1}_{Q \times Q} + 0.8I_{Q \times Q}$, i.e., the standard deviation of each $X$ variable is equal to 1 and the correlation coefficient is 0.2 between each pair of $(X_p, X_q), p \neq q$.

For the function $f$, we choose a non-linear functional form $f(\boldsymbol{X_i}, \epsilon_i, T_i) = 2[X_{i,1}^2 - 0.2I(X_{i,2} > 0)]T_i - 0.8I(X_{i,3} > 0) + 0.8X_{i,4} - 0.4X_{i,5}^2 + \epsilon_i - 3$ for illustration, here $I$ is the indicator function which equals to 1 if the event is true and 0 otherwise. When $\sigma = 1$, the average value of the outcome variable $Y$ is around 0.196, and the average sample uplift in the population is 0.18 with a relatively large standard deviation of 0.312. Note that only a small portion of the $X$ variables are involved in the true data generating process although all of them will be input variables during uplift model training. One could vary the level of $\sigma^2$ to mimic different noise levels and we fix it for now.

As the goal is to check the performance of the proposed confidence interval approach used to compare two pre-trained uplift models, here in the simulation we take model $M_1$ being an XGBoost classification model (Chen & Guestrin, 2016) and the other model $M_2$ being a logistic regression model for illustration. Both models are trained on some dataset to generate uplift predictions given $\boldsymbol{X}$ and essentially serve as ranking functions. It is worth noting that the proposed method is not restricted to compare any specific pre-trained uplift models.

With different random seeds, we could generate simulated full universe data $(\boldsymbol{X}, T, Y)$ under the setting of data generating process. Using each simulated data, we are able to obtain a point estimate of the mean uplift at 5th, 10th, ..., 100th selection percentiles using formula (4), for two pre-trained models $M_1$ and $M_2$ respectively. It is difficult to get an analytical form for the true underlying mean uplift at each selection percentile under this data generating process. Instead we use a Monte Carlo approach and generate the data over a number of different random seeds, for each selection percentile we take the average of the mean uplift point estimate across these simulations, and treat them as the oracle values for mean uplift at different selection percentiles.

We consider balanced treatment allocation first (50% treatment ratio), and 8 different scenarios selection sizes summarized in Table 1 (as percentages of the population size) to illustrate how well the proposed bootstrap confidence intervals cover the oracle mean uplift values at different selection percentiles. Percentages of population size are used here instead of raw counts as we simulate several population sizes within each scenario to demonstrate the bias and standard errors of the proposed point estimate with respect to sample size changes. Additional simulation results for imbalanced treatment allocation is included in Appendix Section A.2.2.

For each scenario, we conduct $K = 200$ simulations with different random seeds. Within each simulation, we perform $B = 100$ outer bootstraps and $D = 10$ inner bootstraps. We only use model $M_1$'s ranking to get the rank-based samples in the two-step sampling scheme, and obtain the inverse inclusion probabilities using the result derived in Appendix Section A.1.1. At each of the 5th, 10th, ..., 100th selection percentile, the coverage probabilities of the constructed 95% bootstrap confidence intervals on the oracle mean uplift value are computed for these 2 models, the differences of the mean uplift between the 2 models are also calculated.

## 4.2 SIMULATION RESULTS

Firstly, for each simulation scenario, empirical results show that the coverage probabilities across 200 simulation runs align well with the nominal level 95% for model 1, model 2 and the mean uplift difference between these two models. In addition, the bias of the proposed bootstrap estimator at each selection percentile is close to 0 and decreases as total population size increases. The standard errors of the bootstrap estimator decrease as population size increases for given selection percentile. For a fixed population size, the bias and standard error metrics also improve as the selection size increases.

Detailed results are included in the Appendix Section A.2.1 with a discussion of additional findings among several different scenarios.

Table 1: Simulation scenario reference table

| scenario id | rank-based selection percent | SRS percent |
|---:|---|---|
| 0 | 5.0% | 1.0% |
| 1 | 10.0% | 5.0% |
| 2 | 5.0% | 0.5% |
| 3 | 10.0% | 1.0% |
| 4 | 5.0% | 5.0% |
| 5 | 1.0% | 0.1% |
| 6 | 5.0% | 10.0% |
| 7 | 1.0% | 10.0% |

## 5 REAL DATA APPLICATION

First, we apply the proposed method to the Criteo Uplift Prediction Dataset, which is released by the Criteo AI Lab (Diemert Eustache, Betlei Artem et al., 2018). This dataset has nearly 14MM rows in total with 12 anonymized feature values. The binary treatment variable shall be independent of the features by design based on the data description in the paper, and the treatment ratio is 85%. There are two binary outcome variables: visit and conversion, and we consider to model the visit outcome variable here as suggested in the paper. The average visit rate is 0.047. This dataset is publicly accessible using the Python scikit-uplift package at `https://www.uplift-modeling.com/en/latest/api/datasets/fetch_criteo.html`, or via Criteo's website at `https://ailab.criteo.com/criteo-uplift-prediction-dataset/` which also has a detailed information page.

For illustration purposes, we take a simple random sample of 2MM rows from this data as the population $\mathbb{P}$. Three classification models are trained in Python using the features along with treatment indicator as model inputs to predict visit.

- Model 1 is trained with an XGBoost classifier, with 100 estimators and maximum tree depth as 3;

- Model 2 is trained with the neural network modeling framework with 10 hidden layers;

- Model 3 is trained with a logistic regression model with default parameters.

Then for each model we can generate the uplift estimates by taking the model prediction differences assuming treatment $T = 1$ and $T = 0$ given feature values.

Here we take a simple random sample with size 200,000 (10% of the population size) and then a rank-based sample of size 200,000 using model 1's ranking. We can obtain an uplift curve estimate using the entire population data (2MM rows), as well as using our proposed nested bootstrap approach.

The estimated uplift curves along with the population data estimate for each model are displayed in Figure 3. Here we use dashed lines to represent the uplift curve estimates from the nested bootstrap approach, with point-wise 95% confidence bands represented by the dotted lines. Each solid lines refers to the uplift curve estimate using the entire population data, which is also just an estimate of the underlying oracle uplift curve rather than the oracle uplift curve itself. The oracle uplift curve is unknown to us in this real data example as we don't have the full knowledge of the underlying data generating process.

To understand the model differences better, we could take a closer look at pair-wise model comparison by checking the estimates of the differences between the uplift curves at each selection percentile. The difference between model 1 and model 2 is displayed in Figure 4. Here we can see that the 95% confidence interval at the 5th selection percentile is above 0, which suggests that model 1 is significantly better than model 2 in terms of cumulative gain at this selection size level. The confidence intervals at other selection percentiles do not seem to indicate a significant difference.

The model difference comparison of model 1 vs. model 3 and model 2 vs. model 3 are included in Figures 7 and 8 in the Appendix Section A.3.1.

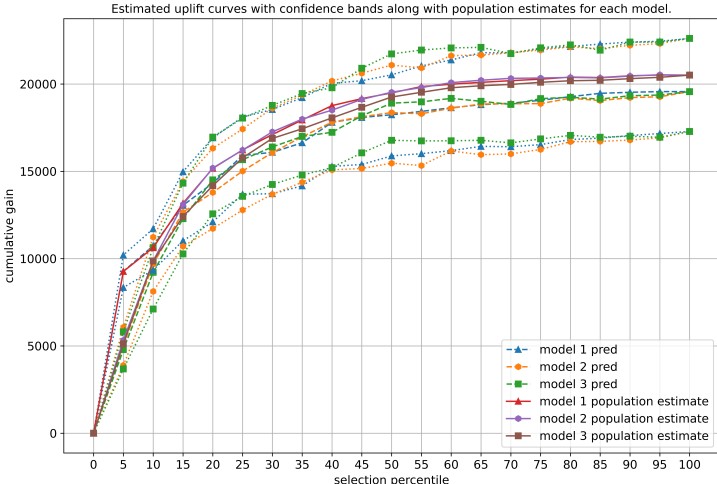

Figure 3: Uplift curve estimates from the nested Bootstrap approach along with the population data estimate for Criteo data.

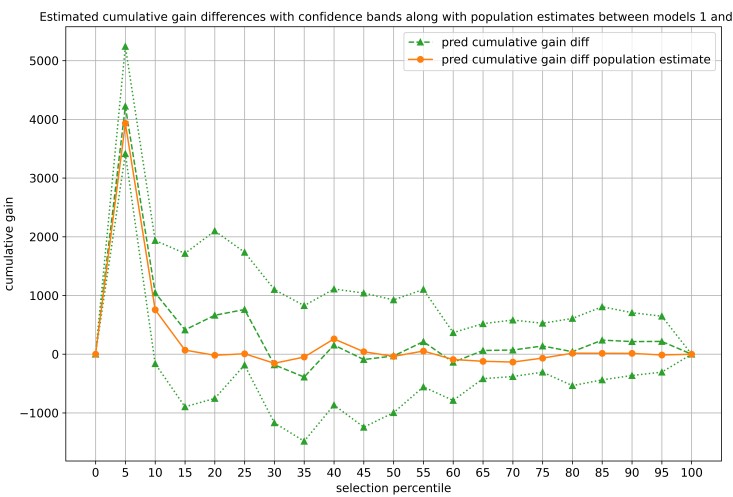

Figure 4: Estimated uplift curve differences between models 1 and 2 for the Criteo data.

In addition to the Criteo data application, we also include another real data application to the Mega-Fon dataset in the Appendix Section A.3, which shows a clear separation of the models being compared.

## 6    DISCUSSION

Through empirical validation using simulation studies, the proposed nested Bootstrap approach is shown to reliably estimate the uplift curve on the entire population and assess its uncertainty with a random subsample. This can be highly relevant and useful in practice for cost sensitive marketing channels such as direct mail, or situations where determining the optimal selection size is critical. In addition, when there are multiple candidate models to be compared with a benchmark model, the proposed approach could generate uplift curve estimates with confidence bands for all models

even when 1 model is used for ranking based selection in production. It saves the practitioner from splitting the selection universe into many segments with insufficient sample sizes for each model, which leads to underpowered comparisons.

For iterative uplift model development in practice, there could be a trade-off between exploration and exploitation when balancing the sample size of the simple random sampling vs. the model ranked based selection given fixed total selection size. Using the proposed 2-step sampling scheme can help collect a simple random subset of the population as future training data compared to purely selecting the top ranked candidates to maximize for immediate uplift gains. It helps mitigate model training data bias due to only including samples with top model ranks and make the model prediction more generalizable on the entire population in the long run. Additionally, one may consider a different sampling scheme than the proposed approach that also collects a random sample from the population and apply the nested bootstrap approach with corresponding inclusion probabilities.

On the computation complexity of the nested bootstrap algorithm, the cost of computing the uplift curve estimate using the population-level data for one model is $\mathcal{O}(N \log(N))$ due to sorting. With $B$ outer bootstraps and $D$ inner bootstraps for total $S$ candidate models, the overall computation complexity would be $\mathcal{O}(N \log(N) BDS)$. Empirically, it took us less than 1 hour to conduct the analysis of the Criteo Data application on a single Amazon AWS r5.24xlarge EC2 instance with no special code optimizations.

Note that if one is interested in the model difference at various selection sizes, some multiplicity control techniques could be applied. For instance, one could leverage the bootstrap samples to construct a simultaneous confidence band at the selection sizes of interest instead of constructing a point-wise confidence band, or use other methods such as Bonferroni adjustment when only a few selection sizes are of interest.

As seen in one of the simulation example, the proposed method may not work very well when sample size is very small as the mean uplift estimates becomes very noisy with limited data from either treatment and control groups. The theoretical properties of the proposed method and its finite population adjustment are interesting future research directions.

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

## A   APPENDIX

### A.1   INCLUSION PROBABILITY FORMULA DERIVATION

#### A.1.1   INCLUSION PROBABILITY FORMULA WITH 1 RANKING MODEL

Following the notations above, the size of the entire selection universe is $N$. When we only use 1 model (denoted as $M_1$) for ranking, based on the sampling scheme introduced in Section 3.1, we take $n_r$ samples randomly from the universe at Step 1, then at Step 2 the top $n - n_r$ samples ranked by model $M_1$ will be selected. We denote model rank as $m_i$ for individual $i$ when ranked by $M_1$. Here a larger value of model rank corresponds to a smaller model prediction in uplift, and the top candidate has rank equal to 1.

It is not difficult to see that when the individual's model rank is greater than $n$, it could only be selected in the random selection step (Step 1) with probability $n_r/N$. When the model rank is smaller than $n - n_r$, the selection probability equals 1 as these candidates will for sure be included in this 2-step procedure.

For individual with model rank in between $n - n_r + 1$ and $n$, let $C_1$ and $C_2$ denote the selected set in Step 1 and 2, respectively. We have $P(i \in C) = P(i \in C_1) + P(i \in C_2)$. For the random selection we have $P(i \in C_1) = n_r/N$. For the event $\{i \in C_2\}$, it is true if and only if at the random selection Step 1, individual $i$ is not selected, and at least $m_i - (n - n_r)$ individuals with rank in between 1 and $(m_i - 1)$ are selected. In this case, there are at most $m_i - 1 - [m_i - (n - n_r)] = n - n_r - 1$ units left after Step 1 with model rank smaller than individual $i$, and individual $i$ will be selected in Step 2.

Therefore we have $P(i \in C_2) = (1 - n_r/N) \sum_{j=m_i-(n-n_r)}^{\min(m_i-1,n_r)} \binom{m_i-1}{j}\binom{N-m_i}{n_r-j}/\binom{N-1}{n_r}$ and the summand here is equal to the probability mass function of a Hyper-geometric random variable with population size equal to $N - 1$, total number of successes as $m_i - 1$, number of draws equal to $n_r$, being evaluated at $j$ successes out of the draws. Thus for an individual $i$ with model rank $m_i$, we have

$$
\begin{aligned}
&P(i \in C) \\
&= \begin{cases}
n_r/N, & \text{if } m_i > n; \\
1, & \text{if } m_i <= n - n_r; \\
n_r/N + (1 - n_r/N) \sum_{j=m_i-(n-n_r)}^{\min(m_i-1,n_r)} \binom{m_i-1}{j}\binom{N-m_i}{n_r-j}/\binom{N-1}{n_r}, & \text{if } n \geq m_i \geq n - n_r + 1.
\end{cases}
\end{aligned}
$$

If we denote $k = m_i - 1 - j$, we have this equivalent expression

$$
P(i \in C_2) = (1 - n_r/N) \sum_{k=\max(0,m_i-1-n_r)}^{n-n_r-1} \binom{m_i-1}{k}\binom{N-m_i}{N-1-n_r-k}/\binom{N-1}{N-1-n_r}
$$

when $n \geq m_i \geq n - n_r + 1$. This can also be seen using the symmetry of the Hyper-geometric distribution, where we consider the probability of observing no more than $n - n_r - 1$ successes from a Hyper-geometric$(N - 1, m_i - 1, N - 1 - n_r)$ random variable.

#### A.1.2   INCLUSION PROBABILITY FORMULA WITH MORE THAN 1 RANKING MODEL

Assume we have in total $S_0$ ranking models, let $m_{i,s}$ represents the model rank of individual $i$ with respect to candidate model $M_s$, $s = 1, ..., S_0$. Based on the sampling scheme introduced in Section 3.1, we take $n_r$ samples randomly from the universe with size $N$ at Step 1 first. At Step 2, the remaining selection universe will be randomly split into $S_0$ sub-universes with sizes $N'_1, N'_2, ..., N'_{S_0}$, and the top $(n - n_r)N'_s/(N - n_r)$ candidates ranked by model $s$ accordingly will be selected. We have $\sum_{s=1}^{S_0} N'_s = N - n_r$, and selected samples in Step 1 and Step 2 will form the selection set $C$ with total size $n$.

Here we use $n'_s = (n - n_r)N'_s/(N - n_r)$ to denote the rank-based selection size from sub-universe $s$ during Step 2 for brevity. For any individual $i$, if $m_{i,s} <= n'_s$ is true for all $s = 1, ..., S_0$, then $P(i \in C) = 1$ as it must be selected in Step 2 unless already selected in Step 1. If $m_{i,s} > N - N'_s + n'_s$ is true for all $s = 1, ..., S_0$, then it could only be selected possibly in Step 1 with probability $n_r/N$.

For other cases, let $C_1$ denote the selected set in Step 1, and $C_{2,s}$ denote the selected set at Step 2 from sub-universe $s$, $s = 1, ..., S_0$. We have $P(i \in C) = P(i \in C_1) + \sum_{s=1}^{S_0} P(i \in C_{2,s})$. As Step 1 is SRS, we have $P(i \in C_1) = n_r/N$.

Now consider the event when individual $i$ is selected in sub-universe $s$ at Step 2, i.e. $\{i \in C_{2,s}\}$. This can be decomposed into the following events: (a) individual $i$ is not selected in Step 1; (b) conditional on the event in (a), individual $i$ is then assigned to sub-universe $s$; and (c): conditional on the events in (a) and (b), individual $i$ gets selected in Step 2.

For the event in (c) to occur, there needs to be no greater than $n'_s - 1$ units in sub-universe $s$ with model rank in between 1 and $m_{i,s} - 1$ with respect to model $M_s$ for individual $i$ to be selected into $C_{2,s}$. It can be expressed as the cumulative distribution function of a Hyper-geometric($N - 1$, $m_{i,s} - 1$, $N'_s - 1$) being evaluated at $n'_s - 1$:

$$
\begin{aligned}
&P(i \in C_{2,s}) \\
&= \begin{cases}
0, & \text{if } m_{i,s} > N - N'_s + n'_s; \\
(1 - n_r/N)\frac{N'_s}{N - n_r}, & \text{if } m_{i,s} <= n'_s; \\
(1 - n_r/N)\frac{N'_s}{N - n_r}P(Z \le n'_s - 1), & \text{if } N - N'_s + n'_s \ge m_{i,s} \ge n'_s + 1,
\end{cases}
\end{aligned}
$$

where $Z \sim$ Hyper-geometric($N - 1, m_{i,s} - 1, N'_s - 1$), and

$$
P(Z \le n'_s - 1) = \sum_{k=\max(0, N'_s - 1 - N + m_{i,s})}^{n'_s - 1} \frac{\binom{m_{i,s}-1}{k}\binom{N'_s - m_{i,s}}{N'_s - 1 - k}}{\binom{N-1}{N'_s - 1}}
$$

.

Finally, we could get $P(i \in C)$ via $P(i \in C) = P(i \in C_1) + \sum_{s=1}^{S_0} P(i \in C_{2,s})$.

## A.2 DETAILED SIMULATION RESULTS

### A.2.1 RESULTS FOR BALANCED TREATMENT ALLOCATION

In this section, we include the detailed simulation results and share some interesting findings. First, note that the scenarios 3 and 7 have the same total number of samples. Scenario 1 has more simple random samples than scenario 3 but same number of rank-based samples. The standard errors at each population size and selection percentile in scenario 7 are smaller than the corresponding ones in scenario 3, as the percentage of SRS is higher in scenario 7. Same phenomenon can be observed when comparing scenarios 1 and 6 as both of them have same total sample size though different split between rank-based samples and simple random samples.

Bias and standard error performance for scenario 1 is nearly uniformly better than scenario 3, which is expected as it has more simple random samples and same rank-based selected samples. Meanwhile interestingly the results in scenario 7 seems better or nearly the same compared to scenario 1 even though scenario 1 has more total selected samples (but fewer simple random samples).

You may notice some under coverage issues at the 5th selection percentile for Scenario 5 in Table 12. Note that under this setting there are very limited simple random samples, the total selected sample size is small as well, which leads to unstable mean uplift estimates. The coverage probabilities are getting closer to the nominal 95% level when selected sample size increases.

Table 2: Coverage probability results for uplift curves in scenario 0 at different population sizes and selection percentiles.

| population size percentile | model 1 | | | model 2 | | | model diff | | |
|---|---|---|---|---|---|---|---|---|---|
| | 200000 | 400000 | 800000 | 200000 | 400000 | 800000 | 200000 | 400000 | 800000 |
| 5 | 0.970 | 0.965 | 0.965 | 0.930 | 0.945 | 0.920 | 0.930 | 0.950 | 0.930 |
| 10 | 0.945 | 0.940 | 0.905 | 0.915 | 0.930 | 0.920 | 0.925 | 0.965 | 0.945 |
| 15 | 0.940 | 0.935 | 0.915 | 0.910 | 0.935 | 0.935 | 0.935 | 0.945 | 0.940 |
| 20 | 0.955 | 0.945 | 0.920 | 0.930 | 0.930 | 0.940 | 0.945 | 0.950 | 0.930 |
| 25 | 0.940 | 0.945 | 0.935 | 0.915 | 0.945 | 0.950 | 0.945 | 0.960 | 0.960 |
| 30 | 0.915 | 0.945 | 0.925 | 0.905 | 0.950 | 0.945 | 0.935 | 0.945 | 0.935 |
| 35 | 0.930 | 0.920 | 0.935 | 0.905 | 0.960 | 0.940 | 0.915 | 0.940 | 0.900 |
| 40 | 0.940 | 0.930 | 0.925 | 0.915 | 0.950 | 0.925 | 0.930 | 0.940 | 0.930 |
| 45 | 0.950 | 0.930 | 0.935 | 0.935 | 0.955 | 0.940 | 0.920 | 0.945 | 0.920 |
| 50 | 0.955 | 0.930 | 0.940 | 0.925 | 0.955 | 0.940 | 0.930 | 0.950 | 0.940 |
| 55 | 0.955 | 0.940 | 0.960 | 0.915 | 0.935 | 0.930 | 0.920 | 0.945 | 0.960 |
| 60 | 0.955 | 0.940 | 0.965 | 0.920 | 0.940 | 0.940 | 0.910 | 0.950 | 0.950 |
| 65 | 0.950 | 0.940 | 0.975 | 0.935 | 0.930 | 0.945 | 0.890 | 0.955 | 0.950 |
| 70 | 0.945 | 0.925 | 0.965 | 0.945 | 0.920 | 0.955 | 0.910 | 0.950 | 0.940 |
| 75 | 0.935 | 0.940 | 0.975 | 0.935 | 0.910 | 0.945 | 0.945 | 0.950 | 0.920 |
| 80 | 0.930 | 0.935 | 0.970 | 0.935 | 0.915 | 0.960 | 0.945 | 0.940 | 0.945 |
| 85 | 0.940 | 0.950 | 0.960 | 0.930 | 0.920 | 0.940 | 0.935 | 0.935 | 0.945 |
| 90 | 0.935 | 0.950 | 0.965 | 0.920 | 0.950 | 0.940 | 0.935 | 0.940 | 0.940 |
| 95 | 0.940 | 0.940 | 0.960 | 0.930 | 0.945 | 0.945 | 0.935 | 0.960 | 0.955 |
| 100 | 0.940 | 0.950 | 0.950 | 0.940 | 0.950 | 0.950 | 1.000 | 1.000 | 1.000 |

Table 3: Performance results of model difference in mean uplift estimator results for scenario 0 at different population sizes and selection percentiles.

| population size percentile | model diff bias (SE) | | |
|---|---|---|---|
| | 200000 | 400000 | 800000 |
| 5 | -0.002(0.089) | -0.007(0.058) | -0.003(0.044) |
| 10 | 0.005(0.063) | -0.005(0.037) | -0.000(0.031) |
| 15 | 0.007(0.051) | -0.002(0.032) | 0.002(0.027) |
| 20 | 0.004(0.044) | -0.002(0.028) | 0.002(0.021) |
| 25 | 0.002(0.039) | -0.002(0.024) | 0.002(0.018) |
| 30 | 0.002(0.034) | -0.004(0.021) | 0.002(0.016) |
| 35 | 0.002(0.030) | -0.003(0.019) | 0.002(0.015) |
| 40 | 0.002(0.025) | -0.002(0.016) | 0.002(0.013) |
| 45 | 0.002(0.021) | -0.001(0.014) | 0.001(0.010) |
| 50 | 0.002(0.018) | -0.001(0.012) | 0.001(0.009) |
| 55 | 0.001(0.015) | -0.000(0.010) | 0.001(0.007) |
| 60 | 0.001(0.014) | -0.000(0.009) | 0.001(0.006) |
| 65 | 0.000(0.012) | -0.000(0.008) | 0.001(0.005) |
| 70 | 0.000(0.010) | 0.000(0.006) | 0.001(0.005) |
| 75 | 0.000(0.008) | -0.000(0.005) | 0.001(0.004) |
| 80 | 0.000(0.007) | -0.000(0.005) | 0.000(0.003) |
| 85 | 0.000(0.006) | -0.001(0.004) | 0.000(0.003) |
| 90 | -0.000(0.004) | -0.001(0.003) | 0.000(0.002) |
| 95 | -0.000(0.003) | -0.000(0.002) | 0.000(0.001) |
| 100 | 0.000(0.000) | 0.000(0.000) | 0.000(0.000) |

A.2.2    SIMULATION RESULTS FOR TREATMENT RATIO EQUAL TO 75%

Table 4: Coverage probability results for uplift curves in scenario 1 at different population sizes and selection percentiles.

| population size percentile | model 1 | | | model 2 | | | model diff | | |
|---|---|---|---|---|---|---|---|---|---|
| | 200000 | 400000 | 800000 | 200000 | 400000 | 800000 | 200000 | 400000 | 800000 |
| 5 | 0.950 | 0.965 | 0.950 | 0.925 | 0.935 | 0.940 | 0.920 | 0.940 | 0.935 |
| 10 | 0.965 | 0.965 | 0.975 | 0.935 | 0.935 | 0.965 | 0.930 | 0.945 | 0.945 |
| 15 | 0.955 | 0.960 | 0.935 | 0.905 | 0.940 | 0.940 | 0.900 | 0.945 | 0.920 |
| 20 | 0.925 | 0.955 | 0.955 | 0.925 | 0.930 | 0.955 | 0.935 | 0.935 | 0.940 |
| 25 | 0.935 | 0.975 | 0.940 | 0.925 | 0.955 | 0.930 | 0.935 | 0.950 | 0.915 |
| 30 | 0.940 | 0.955 | 0.965 | 0.930 | 0.960 | 0.920 | 0.920 | 0.925 | 0.930 |
| 35 | 0.915 | 0.915 | 0.940 | 0.940 | 0.965 | 0.940 | 0.920 | 0.920 | 0.945 |
| 40 | 0.925 | 0.935 | 0.960 | 0.930 | 0.965 | 0.950 | 0.930 | 0.945 | 0.950 |
| 45 | 0.925 | 0.940 | 0.965 | 0.930 | 0.960 | 0.940 | 0.925 | 0.910 | 0.950 |
| 50 | 0.930 | 0.945 | 0.950 | 0.935 | 0.955 | 0.930 | 0.970 | 0.930 | 0.930 |
| 55 | 0.930 | 0.940 | 0.915 | 0.960 | 0.940 | 0.905 | 0.935 | 0.935 | 0.950 |
| 60 | 0.945 | 0.950 | 0.925 | 0.940 | 0.945 | 0.920 | 0.945 | 0.955 | 0.980 |
| 65 | 0.925 | 0.945 | 0.940 | 0.945 | 0.950 | 0.920 | 0.940 | 0.955 | 0.960 |
| 70 | 0.940 | 0.955 | 0.925 | 0.955 | 0.955 | 0.915 | 0.945 | 0.945 | 0.970 |
| 75 | 0.940 | 0.945 | 0.925 | 0.950 | 0.950 | 0.910 | 0.940 | 0.925 | 0.940 |
| 80 | 0.935 | 0.955 | 0.910 | 0.930 | 0.955 | 0.920 | 0.940 | 0.945 | 0.915 |
| 85 | 0.930 | 0.950 | 0.900 | 0.955 | 0.960 | 0.930 | 0.950 | 0.980 | 0.940 |
| 90 | 0.925 | 0.940 | 0.915 | 0.945 | 0.945 | 0.925 | 0.945 | 0.980 | 0.930 |
| 95 | 0.935 | 0.940 | 0.905 | 0.935 | 0.945 | 0.920 | 0.935 | 0.935 | 0.925 |
| 100 | 0.940 | 0.945 | 0.915 | 0.940 | 0.945 | 0.915 | 1.000 | 1.000 | 1.000 |

Table 5: Performance results of model difference in mean uplift estimator results for scenario 1 at different population sizes and selection percentiles.

| population size percentile | model diff bias (SE) | | |
|---|---|---|---|
| | 200000 | 400000 | 800000 |
| 5 | 0.002(0.036) | 0.001(0.024) | -0.001(0.018) |
| 10 | 0.002(0.024) | -0.000(0.016) | 0.000(0.012) |
| 15 | 0.002(0.022) | 0.000(0.014) | 0.001(0.010) |
| 20 | 0.003(0.018) | -0.000(0.013) | 0.000(0.009) |
| 25 | 0.002(0.015) | -0.001(0.011) | 0.000(0.008) |
| 30 | 0.002(0.014) | -0.001(0.010) | 0.001(0.007) |
| 35 | 0.002(0.012) | -0.001(0.009) | 0.001(0.006) |
| 40 | 0.001(0.010) | -0.000(0.007) | 0.000(0.005) |
| 45 | 0.001(0.008) | -0.001(0.006) | 0.000(0.004) |
| 50 | 0.001(0.007) | -0.000(0.005) | 0.000(0.004) |
| 55 | -0.000(0.006) | -0.000(0.004) | 0.000(0.003) |
| 60 | -0.000(0.005) | -0.000(0.004) | 0.000(0.002) |
| 65 | -0.000(0.005) | -0.000(0.003) | 0.000(0.002) |
| 70 | -0.000(0.004) | -0.000(0.003) | 0.000(0.002) |
| 75 | 0.000(0.003) | -0.000(0.002) | 0.000(0.002) |
| 80 | -0.000(0.003) | -0.000(0.002) | 0.000(0.001) |
| 85 | 0.000(0.002) | -0.000(0.001) | 0.000(0.001) |
| 90 | 0.000(0.002) | -0.000(0.001) | -0.000(0.001) |
| 95 | 0.000(0.001) | -0.000(0.001) | 0.000(0.001) |
| 100 | 0.000(0.000) | 0.000(0.000) | 0.000(0.000) |

Table 6: Coverage probability results for uplift curves in scenario 2 at different population sizes and selection percentiles.

| population size percentile | model 1 | | | model 2 | | | model diff | | |
|---|---|---|---|---|---|---|---|---|---|
| | 200000 | 400000 | 800000 | 200000 | 400000 | 800000 | 200000 | 400000 | 800000 |
| 5 | 0.950 | 0.965 | 0.975 | 0.940 | 0.955 | 0.925 | 0.935 | 0.955 | 0.925 |
| 10 | 0.900 | 0.940 | 0.940 | 0.925 | 0.945 | 0.900 | 0.920 | 0.955 | 0.910 |
| 15 | 0.935 | 0.960 | 0.930 | 0.925 | 0.920 | 0.925 | 0.925 | 0.955 | 0.905 |
| 20 | 0.925 | 0.940 | 0.930 | 0.940 | 0.930 | 0.935 | 0.955 | 0.965 | 0.890 |
| 25 | 0.910 | 0.930 | 0.930 | 0.945 | 0.930 | 0.930 | 0.945 | 0.970 | 0.905 |
| 30 | 0.900 | 0.945 | 0.930 | 0.920 | 0.915 | 0.940 | 0.910 | 0.925 | 0.915 |
| 35 | 0.900 | 0.930 | 0.925 | 0.925 | 0.945 | 0.920 | 0.905 | 0.970 | 0.890 |
| 40 | 0.905 | 0.920 | 0.930 | 0.935 | 0.940 | 0.945 | 0.905 | 0.950 | 0.920 |
| 45 | 0.890 | 0.955 | 0.925 | 0.920 | 0.940 | 0.940 | 0.940 | 0.955 | 0.920 |
| 50 | 0.885 | 0.950 | 0.945 | 0.925 | 0.935 | 0.945 | 0.925 | 0.930 | 0.920 |
| 55 | 0.895 | 0.930 | 0.930 | 0.930 | 0.935 | 0.940 | 0.930 | 0.955 | 0.935 |
| 60 | 0.905 | 0.920 | 0.935 | 0.905 | 0.935 | 0.960 | 0.925 | 0.920 | 0.940 |
| 65 | 0.900 | 0.920 | 0.940 | 0.905 | 0.920 | 0.960 | 0.920 | 0.935 | 0.950 |
| 70 | 0.905 | 0.915 | 0.955 | 0.915 | 0.925 | 0.955 | 0.925 | 0.925 | 0.955 |
| 75 | 0.920 | 0.915 | 0.955 | 0.935 | 0.925 | 0.950 | 0.950 | 0.910 | 0.960 |
| 80 | 0.920 | 0.915 | 0.950 | 0.930 | 0.915 | 0.970 | 0.965 | 0.925 | 0.960 |
| 85 | 0.925 | 0.905 | 0.945 | 0.925 | 0.930 | 0.960 | 0.930 | 0.925 | 0.955 |
| 90 | 0.935 | 0.925 | 0.965 | 0.935 | 0.920 | 0.945 | 0.950 | 0.945 | 0.965 |
| 95 | 0.930 | 0.925 | 0.960 | 0.920 | 0.935 | 0.950 | 0.940 | 0.940 | 0.950 |
| 100 | 0.925 | 0.925 | 0.950 | 0.925 | 0.925 | 0.950 | 1.000 | 1.000 | 1.000 |

Table 7: Performance results of model difference in mean uplift estimator results for scenario 2 at different population sizes and selection percentiles.

| population size percentile | model diff bias (SE) | | |
|---|---|---|---|
| | 200000 | 400000 | 800000 |
| 5 | -0.019(0.117) | -0.008(0.077) | 0.003(0.062) |
| 10 | -0.003(0.087) | -0.009(0.053) | 0.006(0.046) |
| 15 | 0.001(0.072) | -0.008(0.047) | 0.006(0.039) |
| 20 | 0.004(0.063) | -0.008(0.042) | 0.004(0.034) |
| 25 | 0.006(0.054) | -0.005(0.036) | 0.003(0.028) |
| 30 | 0.006(0.047) | -0.006(0.031) | 0.003(0.025) |
| 35 | 0.007(0.042) | -0.004(0.026) | 0.003(0.022) |
| 40 | 0.005(0.036) | -0.003(0.024) | 0.002(0.018) |
| 45 | 0.005(0.031) | -0.002(0.020) | 0.001(0.015) |
| 50 | 0.005(0.026) | -0.001(0.017) | 0.002(0.013) |
| 55 | 0.004(0.022) | -0.000(0.015) | 0.002(0.011) |
| 60 | 0.003(0.020) | -0.001(0.013) | 0.001(0.009) |
| 65 | 0.002(0.017) | -0.000(0.012) | 0.001(0.007) |
| 70 | 0.002(0.014) | -0.000(0.010) | 0.001(0.006) |
| 75 | 0.002(0.011) | -0.000(0.008) | 0.001(0.005) |
| 80 | 0.001(0.009) | -0.000(0.007) | 0.000(0.004) |
| 85 | 0.001(0.008) | -0.001(0.005) | 0.000(0.004) |
| 90 | 0.000(0.006) | -0.001(0.004) | 0.000(0.003) |
| 95 | 0.000(0.004) | -0.000(0.003) | 0.000(0.002) |
| 100 | 0.000(0.000) | 0.000(0.000) | 0.000(0.000) |

Table 8: Coverage probability results for uplift curves in scenario 3 at different population sizes and selection percentiles.

| population size percentile | model 1 | | | model 2 | | | model diff | | |
|---|---|---|---|---|---|---|---|---|---|
| | 200000 | 400000 | 800000 | 200000 | 400000 | 800000 | 200000 | 400000 | 800000 |
| 5 | 0.915 | 0.970 | 0.950 | 0.915 | 0.935 | 0.920 | 0.915 | 0.940 | 0.910 |
| 10 | 0.995 | 0.995 | 0.990 | 0.910 | 0.925 | 0.940 | 0.900 | 0.935 | 0.945 |
| 15 | 0.950 | 0.930 | 0.955 | 0.925 | 0.920 | 0.935 | 0.950 | 0.935 | 0.945 |
| 20 | 0.940 | 0.960 | 0.945 | 0.935 | 0.945 | 0.950 | 0.960 | 0.965 | 0.940 |
| 25 | 0.940 | 0.970 | 0.930 | 0.910 | 0.925 | 0.955 | 0.935 | 0.960 | 0.975 |
| 30 | 0.930 | 0.950 | 0.940 | 0.915 | 0.940 | 0.935 | 0.925 | 0.965 | 0.945 |
| 35 | 0.945 | 0.945 | 0.930 | 0.885 | 0.970 | 0.920 | 0.920 | 0.955 | 0.925 |
| 40 | 0.925 | 0.955 | 0.930 | 0.935 | 0.960 | 0.920 | 0.920 | 0.930 | 0.920 |
| 45 | 0.945 | 0.970 | 0.940 | 0.940 | 0.945 | 0.930 | 0.940 | 0.920 | 0.940 |
| 50 | 0.955 | 0.950 | 0.940 | 0.965 | 0.960 | 0.925 | 0.920 | 0.935 | 0.945 |
| 55 | 0.960 | 0.945 | 0.935 | 0.935 | 0.950 | 0.930 | 0.930 | 0.950 | 0.975 |
| 60 | 0.960 | 0.950 | 0.955 | 0.950 | 0.945 | 0.940 | 0.925 | 0.975 | 0.950 |
| 65 | 0.920 | 0.950 | 0.950 | 0.935 | 0.935 | 0.940 | 0.915 | 0.955 | 0.920 |
| 70 | 0.930 | 0.950 | 0.955 | 0.940 | 0.940 | 0.935 | 0.925 | 0.945 | 0.940 |
| 75 | 0.920 | 0.945 | 0.950 | 0.940 | 0.950 | 0.930 | 0.955 | 0.950 | 0.940 |
| 80 | 0.920 | 0.955 | 0.950 | 0.930 | 0.950 | 0.950 | 0.950 | 0.940 | 0.950 |
| 85 | 0.905 | 0.965 | 0.940 | 0.920 | 0.955 | 0.930 | 0.940 | 0.955 | 0.945 |
| 90 | 0.920 | 0.950 | 0.920 | 0.930 | 0.965 | 0.920 | 0.955 | 0.910 | 0.915 |
| 95 | 0.920 | 0.950 | 0.925 | 0.915 | 0.955 | 0.920 | 0.960 | 0.945 | 0.970 |
| 100 | 0.920 | 0.955 | 0.935 | 0.920 | 0.955 | 0.935 | 1.000 | 1.000 | 1.000 |

Table 9: Performance results of model difference in mean uplift estimator for scenario 3 at different population sizes and selection percentiles.

| population size percentile | model diff bias (SE) | | |
|---|---|---|---|
| | 200000 | 400000 | 800000 |
| 5 | -0.007(0.086) | -0.006(0.054) | -0.006(0.042) |
| 10 | -0.000(0.056) | -0.005(0.037) | -0.004(0.028) |
| 15 | 0.004(0.044) | -0.002(0.031) | -0.000(0.024) |
| 20 | 0.005(0.040) | -0.003(0.027) | -0.000(0.020) |
| 25 | 0.004(0.035) | -0.002(0.024) | 0.000(0.017) |
| 30 | 0.003(0.031) | -0.003(0.020) | 0.001(0.015) |
| 35 | 0.003(0.027) | -0.002(0.017) | 0.001(0.013) |
| 40 | 0.002(0.023) | -0.002(0.014) | 0.001(0.011) |
| 45 | 0.002(0.019) | -0.001(0.013) | 0.001(0.009) |
| 50 | 0.002(0.017) | -0.000(0.011) | 0.001(0.008) |
| 55 | 0.003(0.014) | 0.000(0.009) | 0.001(0.007) |
| 60 | 0.002(0.012) | 0.000(0.008) | 0.000(0.006) |
| 65 | 0.002(0.011) | -0.000(0.007) | 0.000(0.005) |
| 70 | 0.001(0.009) | 0.000(0.006) | 0.001(0.004) |
| 75 | 0.001(0.007) | -0.000(0.005) | 0.000(0.004) |
| 80 | 0.001(0.006) | -0.000(0.004) | 0.000(0.003) |
| 85 | 0.000(0.005) | -0.001(0.003) | 0.000(0.002) |
| 90 | -0.000(0.004) | -0.000(0.003) | 0.000(0.002) |
| 95 | 0.000(0.003) | -0.000(0.002) | 0.000(0.001) |
| 100 | 0.000(0.000) | 0.000(0.000) | 0.000(0.000) |

Table 10: Coverage probability results for uplift curves in scenario 4 at different population sizes and selection percentiles.

| population size percentile | model 1 | | | model 2 | | | model diff | | |
|---|---|---|---|---|---|---|---|---|---|
| | 200000 | 400000 | 800000 | 200000 | 400000 | 800000 | 200000 | 400000 | 800000 |
| 5 | 0.945 | 0.975 | 0.960 | 0.950 | 0.940 | 0.930 | 0.950 | 0.950 | 0.930 |
| 10 | 0.940 | 0.920 | 0.925 | 0.945 | 0.940 | 0.965 | 0.935 | 0.945 | 0.945 |
| 15 | 0.940 | 0.970 | 0.935 | 0.915 | 0.955 | 0.975 | 0.930 | 0.950 | 0.940 |
| 20 | 0.945 | 0.955 | 0.960 | 0.930 | 0.945 | 0.970 | 0.920 | 0.950 | 0.935 |
| 25 | 0.935 | 0.970 | 0.940 | 0.915 | 0.925 | 0.960 | 0.955 | 0.940 | 0.975 |
| 30 | 0.960 | 0.955 | 0.950 | 0.920 | 0.950 | 0.950 | 0.940 | 0.930 | 0.945 |
| 35 | 0.940 | 0.925 | 0.925 | 0.940 | 0.955 | 0.930 | 0.925 | 0.925 | 0.940 |
| 40 | 0.955 | 0.950 | 0.930 | 0.945 | 0.960 | 0.930 | 0.925 | 0.940 | 0.950 |
| 45 | 0.950 | 0.955 | 0.955 | 0.955 | 0.945 | 0.930 | 0.940 | 0.925 | 0.935 |
| 50 | 0.955 | 0.960 | 0.940 | 0.950 | 0.945 | 0.915 | 0.945 | 0.940 | 0.920 |
| 55 | 0.955 | 0.950 | 0.925 | 0.940 | 0.950 | 0.920 | 0.940 | 0.935 | 0.935 |
| 60 | 0.945 | 0.955 | 0.930 | 0.940 | 0.960 | 0.935 | 0.925 | 0.930 | 0.960 |
| 65 | 0.940 | 0.960 | 0.930 | 0.940 | 0.950 | 0.920 | 0.945 | 0.945 | 0.965 |
| 70 | 0.955 | 0.950 | 0.920 | 0.935 | 0.960 | 0.920 | 0.935 | 0.940 | 0.960 |
| 75 | 0.955 | 0.960 | 0.910 | 0.945 | 0.960 | 0.920 | 0.935 | 0.940 | 0.945 |
| 80 | 0.955 | 0.950 | 0.905 | 0.950 | 0.960 | 0.940 | 0.950 | 0.940 | 0.950 |
| 85 | 0.955 | 0.950 | 0.915 | 0.955 | 0.955 | 0.920 | 0.935 | 0.955 | 0.940 |
| 90 | 0.945 | 0.955 | 0.920 | 0.940 | 0.945 | 0.935 | 0.945 | 0.970 | 0.920 |
| 95 | 0.945 | 0.955 | 0.920 | 0.960 | 0.940 | 0.940 | 0.935 | 0.940 | 0.940 |
| 100 | 0.960 | 0.950 | 0.935 | 0.960 | 0.950 | 0.935 | 1.000 | 1.000 | 1.000 |

Table 11: Performance results of model difference in mean uplift estimator results for scenario 4 at different population sizes and selection percentiles.

| population size percentile | model diff bias (SE) | | |
|---|---|---|---|
| | 200000 | 400000 | 800000 |
| 5 | 0.004(0.037) | 0.001(0.027) | -0.000(0.019) |
| 10 | 0.004(0.028) | 0.000(0.017) | 0.001(0.013) |
| 15 | 0.004(0.024) | 0.000(0.015) | 0.001(0.011) |
| 20 | 0.004(0.020) | -0.000(0.013) | 0.001(0.009) |
| 25 | 0.003(0.016) | -0.001(0.012) | 0.000(0.008) |
| 30 | 0.002(0.014) | -0.001(0.010) | 0.001(0.007) |
| 35 | 0.002(0.013) | -0.001(0.009) | 0.001(0.006) |
| 40 | 0.001(0.010) | -0.001(0.008) | 0.000(0.005) |
| 45 | 0.001(0.009) | -0.001(0.007) | 0.000(0.004) |
| 50 | 0.001(0.008) | -0.000(0.005) | 0.000(0.004) |
| 55 | -0.000(0.006) | -0.000(0.005) | 0.000(0.003) |
| 60 | -0.000(0.006) | -0.000(0.004) | -0.000(0.003) |
| 65 | -0.000(0.005) | -0.000(0.003) | 0.000(0.002) |
| 70 | -0.000(0.004) | -0.000(0.003) | 0.000(0.002) |
| 75 | 0.000(0.004) | -0.000(0.002) | 0.000(0.002) |
| 80 | -0.000(0.003) | -0.000(0.002) | -0.000(0.001) |
| 85 | 0.000(0.002) | -0.000(0.002) | 0.000(0.001) |
| 90 | 0.000(0.002) | -0.000(0.001) | -0.000(0.001) |
| 95 | -0.000(0.001) | -0.000(0.001) | 0.000(0.001) |
| 100 | 0.000(0.000) | 0.000(0.000) | 0.000(0.000) |

Table 12: Coverage probability results for uplift curves in scenario 5 at different population sizes and selection percentiles.

| population size percentile | model 1 200000 | model 1 400000 | model 1 800000 | model 2 200000 | model 2 400000 | model 2 800000 | model diff 200000 | model diff 400000 | model diff 800000 |
|---|---|---|---|---|---|---|---|---|---|
| 5 | 0.510 | 0.710 | 0.795 | 0.660 | 0.925 | 0.950 | 0.680 | 0.935 | 0.955 |
| 10 | 0.875 | 0.915 | 0.935 | 0.935 | 0.940 | 0.935 | 0.945 | 0.955 | 0.930 |
| 15 | 0.940 | 0.950 | 0.960 | 0.935 | 0.950 | 0.950 | 0.955 | 0.950 | 0.945 |
| 20 | 0.950 | 0.930 | 0.945 | 0.945 | 0.935 | 0.950 | 0.970 | 0.920 | 0.955 |
| 25 | 0.945 | 0.940 | 0.950 | 0.950 | 0.935 | 0.955 | 0.945 | 0.945 | 0.945 |
| 30 | 0.945 | 0.935 | 0.975 | 0.935 | 0.920 | 0.940 | 0.945 | 0.960 | 0.935 |
| 35 | 0.945 | 0.930 | 0.945 | 0.945 | 0.925 | 0.930 | 0.950 | 0.960 | 0.940 |
| 40 | 0.940 | 0.930 | 0.945 | 0.945 | 0.905 | 0.940 | 0.940 | 0.960 | 0.965 |
| 45 | 0.935 | 0.920 | 0.950 | 0.935 | 0.900 | 0.955 | 0.965 | 0.950 | 0.960 |
| 50 | 0.955 | 0.900 | 0.925 | 0.930 | 0.900 | 0.965 | 0.950 | 0.970 | 0.950 |
| 55 | 0.940 | 0.900 | 0.920 | 0.930 | 0.895 | 0.950 | 0.945 | 0.955 | 0.935 |
| 60 | 0.940 | 0.900 | 0.935 | 0.935 | 0.900 | 0.950 | 0.935 | 0.925 | 0.915 |
| 65 | 0.925 | 0.915 | 0.945 | 0.930 | 0.900 | 0.950 | 0.940 | 0.940 | 0.925 |
| 70 | 0.925 | 0.925 | 0.925 | 0.910 | 0.905 | 0.955 | 0.935 | 0.930 | 0.925 |
| 75 | 0.935 | 0.915 | 0.915 | 0.915 | 0.910 | 0.945 | 0.935 | 0.940 | 0.945 |
| 80 | 0.930 | 0.915 | 0.935 | 0.905 | 0.910 | 0.935 | 0.950 | 0.925 | 0.950 |
| 85 | 0.930 | 0.885 | 0.920 | 0.920 | 0.910 | 0.925 | 0.960 | 0.940 | 0.930 |
| 90 | 0.915 | 0.895 | 0.915 | 0.910 | 0.910 | 0.930 | 0.975 | 0.945 | 0.920 |
| 95 | 0.920 | 0.895 | 0.925 | 0.920 | 0.910 | 0.940 | 0.975 | 0.940 | 0.920 |
| 100 | 0.920 | 0.910 | 0.925 | 0.920 | 0.910 | 0.925 | 1.000 | 1.000 | 1.000 |

Table 13: Performance results of model difference in mean uplift estimator results for scenario 5 at different population sizes and selection percentiles.

| population size percentile | model diff bias (SE) 200000 | 400000 | 800000 |
|---|---|---|---|
| 5 | -0.052(0.286) | -0.007(0.192) | 0.003(0.137) |
| 10 | -0.001(0.201) | -0.018(0.138) | 0.013(0.098) |
| 15 | 0.009(0.175) | -0.011(0.115) | 0.006(0.082) |
| 20 | 0.013(0.156) | -0.007(0.101) | 0.003(0.069) |
| 25 | 0.012(0.136) | -0.005(0.082) | 0.007(0.060) |
| 30 | 0.010(0.117) | -0.006(0.070) | 0.005(0.053) |
| 35 | 0.009(0.100) | -0.003(0.062) | 0.005(0.045) |
| 40 | 0.007(0.083) | -0.003(0.054) | 0.004(0.039) |
| 45 | 0.006(0.072) | -0.001(0.048) | 0.006(0.034) |
| 50 | 0.006(0.061) | 0.001(0.041) | 0.007(0.029) |
| 55 | 0.007(0.053) | 0.002(0.037) | 0.006(0.026) |
| 60 | 0.008(0.048) | 0.002(0.032) | 0.005(0.022) |
| 65 | 0.006(0.041) | 0.001(0.027) | 0.003(0.019) |
| 70 | 0.005(0.035) | 0.000(0.023) | 0.002(0.016) |
| 75 | 0.005(0.030) | -0.000(0.019) | 0.002(0.013) |
| 80 | 0.004(0.023) | -0.001(0.016) | 0.001(0.011) |
| 85 | 0.003(0.018) | -0.001(0.013) | 0.000(0.009) |
| 90 | 0.002(0.014) | -0.001(0.010) | 0.000(0.007) |
| 95 | 0.001(0.008) | -0.000(0.006) | 0.000(0.005) |
| 100 | 0.000(0.000) | 0.000(0.000) | 0.000(0.000) |

Table 14: Coverage probability results for uplift curves in scenario 6 at different population sizes and selection percentiles.

| population size percentile | model 1 200000 | model 1 400000 | model 1 800000 | model 2 200000 | model 2 400000 | model 2 800000 | model diff 200000 | model diff 400000 | model diff 800000 |
|---|---|---|---|---|---|---|---|---|---|
| 5 | 0.935 | 0.965 | 0.945 | 0.940 | 0.935 | 0.930 | 0.950 | 0.930 | 0.935 |
| 10 | 0.925 | 0.950 | 0.950 | 0.945 | 0.930 | 0.940 | 0.940 | 0.950 | 0.945 |
| 15 | 0.960 | 0.960 | 0.960 | 0.930 | 0.915 | 0.970 | 0.930 | 0.905 | 0.955 |
| 20 | 0.930 | 0.940 | 0.955 | 0.930 | 0.925 | 0.960 | 0.930 | 0.930 | 0.980 |
| 25 | 0.930 | 0.930 | 0.960 | 0.935 | 0.900 | 0.960 | 0.925 | 0.930 | 0.965 |
| 30 | 0.945 | 0.930 | 0.940 | 0.950 | 0.910 | 0.925 | 0.950 | 0.925 | 0.935 |
| 35 | 0.940 | 0.925 | 0.950 | 0.950 | 0.945 | 0.940 | 0.920 | 0.945 | 0.960 |
| 40 | 0.950 | 0.940 | 0.940 | 0.940 | 0.930 | 0.945 | 0.940 | 0.910 | 0.935 |
| 45 | 0.925 | 0.950 | 0.945 | 0.950 | 0.920 | 0.945 | 0.925 | 0.925 | 0.945 |
| 50 | 0.945 | 0.960 | 0.930 | 0.940 | 0.940 | 0.925 | 0.950 | 0.930 | 0.935 |
| 55 | 0.955 | 0.945 | 0.905 | 0.960 | 0.925 | 0.950 | 0.950 | 0.935 | 0.945 |
| 60 | 0.980 | 0.940 | 0.925 | 0.960 | 0.915 | 0.935 | 0.950 | 0.895 | 0.940 |
| 65 | 0.960 | 0.950 | 0.930 | 0.955 | 0.920 | 0.955 | 0.925 | 0.915 | 0.940 |
| 70 | 0.965 | 0.940 | 0.935 | 0.975 | 0.935 | 0.945 | 0.945 | 0.935 | 0.960 |
| 75 | 0.960 | 0.940 | 0.935 | 0.975 | 0.945 | 0.955 | 0.935 | 0.905 | 0.950 |
| 80 | 0.980 | 0.945 | 0.935 | 0.970 | 0.945 | 0.950 | 0.925 | 0.940 | 0.950 |
| 85 | 0.960 | 0.955 | 0.930 | 0.975 | 0.955 | 0.955 | 0.965 | 0.930 | 0.940 |
| 90 | 0.955 | 0.935 | 0.930 | 0.970 | 0.945 | 0.950 | 0.965 | 0.965 | 0.930 |
| 95 | 0.970 | 0.950 | 0.930 | 0.965 | 0.945 | 0.935 | 0.945 | 0.945 | 0.960 |
| 100 | 0.970 | 0.940 | 0.935 | 0.970 | 0.940 | 0.935 | 1.000 | 1.000 | 1.000 |

Table 15: Performance results of model difference in mean uplift estimator results for scenario 6 at different population sizes and selection percentiles.

| population size percentile | model diff bias (SE) 200000 | model diff bias (SE) 400000 | model diff bias (SE) 800000 |
|---|---|---|---|
| 5 | 0.002(0.025) | 0.003(0.019) | -0.001(0.014) |
| 10 | 0.001(0.019) | 0.001(0.013) | -0.001(0.009) |
| 15 | 0.002(0.015) | 0.001(0.012) | -0.000(0.007) |
| 20 | 0.002(0.014) | -0.000(0.010) | -0.000(0.006) |
| 25 | 0.001(0.012) | -0.001(0.008) | -0.000(0.005) |
| 30 | 0.001(0.010) | -0.001(0.007) | -0.000(0.005) |
| 35 | 0.001(0.009) | -0.001(0.006) | 0.000(0.004) |
| 40 | 0.001(0.007) | -0.000(0.006) | -0.000(0.004) |
| 45 | 0.001(0.006) | -0.000(0.005) | -0.000(0.003) |
| 50 | 0.000(0.005) | -0.000(0.004) | 0.000(0.003) |
| 55 | -0.000(0.004) | -0.000(0.004) | 0.000(0.002) |
| 60 | -0.000(0.004) | -0.000(0.003) | 0.000(0.002) |
| 65 | -0.000(0.004) | -0.000(0.003) | 0.000(0.002) |
| 70 | 0.000(0.003) | -0.000(0.002) | 0.000(0.001) |
| 75 | -0.000(0.002) | -0.000(0.002) | 0.000(0.001) |
| 80 | 0.000(0.002) | -0.000(0.001) | 0.000(0.001) |
| 85 | 0.000(0.002) | -0.000(0.001) | 0.000(0.001) |
| 90 | 0.000(0.001) | 0.000(0.001) | 0.000(0.001) |
| 95 | 0.000(0.001) | 0.000(0.001) | 0.000(0.000) |
| 100 | 0.000(0.000) | 0.000(0.000) | 0.000(0.000) |

Table 16: Coverage probability results for uplift curves in scenario 7 at different population sizes and selection percentiles.

| population size percentile | model 1 | | | model 2 | | | model diff | | |
|---|---|---|---|---|---|---|---|---|---|
| | 200000 | 400000 | 800000 | 200000 | 400000 | 800000 | 200000 | 400000 | 800000 |
| 5 | 0.905 | 0.955 | 0.950 | 0.935 | 0.900 | 0.925 | 0.935 | 0.915 | 0.920 |
| 10 | 0.945 | 0.955 | 0.960 | 0.930 | 0.925 | 0.925 | 0.930 | 0.920 | 0.935 |
| 15 | 0.950 | 0.945 | 0.965 | 0.955 | 0.910 | 0.955 | 0.940 | 0.910 | 0.940 |
| 20 | 0.945 | 0.940 | 0.960 | 0.945 | 0.915 | 0.960 | 0.920 | 0.920 | 0.965 |
| 25 | 0.955 | 0.940 | 0.945 | 0.935 | 0.920 | 0.955 | 0.930 | 0.930 | 0.965 |
| 30 | 0.965 | 0.935 | 0.950 | 0.950 | 0.925 | 0.945 | 0.975 | 0.925 | 0.965 |
| 35 | 0.955 | 0.945 | 0.965 | 0.955 | 0.950 | 0.945 | 0.955 | 0.925 | 0.965 |
| 40 | 0.960 | 0.935 | 0.940 | 0.955 | 0.940 | 0.935 | 0.970 | 0.885 | 0.935 |
| 45 | 0.950 | 0.945 | 0.945 | 0.955 | 0.935 | 0.955 | 0.950 | 0.925 | 0.950 |
| 50 | 0.945 | 0.955 | 0.945 | 0.955 | 0.950 | 0.940 | 0.935 | 0.925 | 0.945 |
| 55 | 0.960 | 0.955 | 0.925 | 0.965 | 0.945 | 0.955 | 0.940 | 0.930 | 0.935 |
| 60 | 0.950 | 0.935 | 0.930 | 0.960 | 0.940 | 0.945 | 0.960 | 0.915 | 0.945 |
| 65 | 0.960 | 0.960 | 0.930 | 0.975 | 0.960 | 0.925 | 0.920 | 0.935 | 0.925 |
| 70 | 0.965 | 0.940 | 0.925 | 0.945 | 0.960 | 0.930 | 0.930 | 0.930 | 0.970 |
| 75 | 0.960 | 0.945 | 0.935 | 0.965 | 0.950 | 0.940 | 0.955 | 0.900 | 0.955 |
| 80 | 0.970 | 0.950 | 0.930 | 0.950 | 0.960 | 0.945 | 0.955 | 0.920 | 0.940 |
| 85 | 0.965 | 0.950 | 0.935 | 0.960 | 0.955 | 0.945 | 0.955 | 0.955 | 0.940 |
| 90 | 0.955 | 0.950 | 0.925 | 0.950 | 0.945 | 0.955 | 0.940 | 0.970 | 0.945 |
| 95 | 0.965 | 0.945 | 0.930 | 0.965 | 0.945 | 0.950 | 0.940 | 0.930 | 0.950 |
| 100 | 0.965 | 0.955 | 0.935 | 0.965 | 0.955 | 0.935 | 1.000 | 1.000 | 1.000 |

Table 17: Performance results of model difference in mean uplift estimator results for scenario 7 at different population sizes and selection percentiles.

| population size percentile | model diff bias (SE) | | |
|---|---|---|---|
| | 200000 | 400000 | 800000 |
| 5 | 0.004(0.030) | 0.003(0.021) | -0.001(0.015) |
| 10 | 0.002(0.021) | 0.000(0.015) | -0.000(0.010) |
| 15 | 0.002(0.017) | 0.000(0.012) | -0.000(0.008) |
| 20 | 0.002(0.015) | -0.000(0.011) | -0.000(0.007) |
| 25 | 0.001(0.012) | -0.001(0.009) | -0.000(0.006) |
| 30 | 0.001(0.010) | -0.001(0.008) | -0.000(0.005) |
| 35 | 0.001(0.009) | -0.000(0.007) | 0.000(0.004) |
| 40 | 0.001(0.007) | -0.000(0.006) | -0.000(0.004) |
| 45 | 0.001(0.006) | -0.000(0.005) | -0.000(0.003) |
| 50 | 0.000(0.006) | -0.000(0.005) | 0.000(0.003) |
| 55 | -0.000(0.005) | -0.000(0.004) | 0.000(0.002) |
| 60 | -0.000(0.004) | -0.000(0.003) | 0.000(0.002) |
| 65 | -0.000(0.004) | -0.000(0.003) | 0.000(0.002) |
| 70 | 0.000(0.003) | 0.000(0.002) | 0.000(0.002) |
| 75 | -0.000(0.003) | -0.000(0.002) | 0.000(0.001) |
| 80 | 0.000(0.002) | -0.000(0.002) | 0.000(0.001) |
| 85 | 0.000(0.002) | -0.000(0.001) | 0.000(0.001) |
| 90 | 0.000(0.001) | 0.000(0.001) | 0.000(0.001) |
| 95 | -0.000(0.001) | 0.000(0.001) | 0.000(0.000) |
| 100 | 0.000(0.000) | 0.000(0.000) | 0.000(0.000) |

Table 18: Coverage probability results for scenario 0 at different population sizes with treatment ratio equal to 75%.

| population size percentile | model 1 | | | model 2 | | | model diff | | |
|---|---|---|---|---|---|---|---|---|---|
| | 200000 | 400000 | 800000 | 200000 | 400000 | 800000 | 200000 | 400000 | 800000 |
| 5 | 0.940 | 0.970 | 0.965 | 0.910 | 0.950 | 0.915 | 0.910 | 0.950 | 0.915 |
| 10 | 0.900 | 0.920 | 0.945 | 0.910 | 0.935 | 0.925 | 0.925 | 0.950 | 0.925 |
| 15 | 0.920 | 0.940 | 0.890 | 0.915 | 0.945 | 0.920 | 0.950 | 0.940 | 0.915 |
| 20 | 0.930 | 0.930 | 0.895 | 0.910 | 0.935 | 0.935 | 0.925 | 0.905 | 0.930 |
| 25 | 0.945 | 0.945 | 0.925 | 0.905 | 0.930 | 0.960 | 0.910 | 0.925 | 0.940 |
| 30 | 0.920 | 0.930 | 0.945 | 0.910 | 0.940 | 0.960 | 0.900 | 0.945 | 0.930 |
| 35 | 0.905 | 0.930 | 0.955 | 0.905 | 0.945 | 0.975 | 0.940 | 0.955 | 0.920 |
| 40 | 0.925 | 0.940 | 0.945 | 0.910 | 0.955 | 0.970 | 0.920 | 0.965 | 0.920 |
| 45 | 0.915 | 0.950 | 0.950 | 0.905 | 0.965 | 0.950 | 0.930 | 0.935 | 0.925 |
| 50 | 0.920 | 0.925 | 0.975 | 0.940 | 0.965 | 0.955 | 0.910 | 0.930 | 0.920 |
| 55 | 0.935 | 0.935 | 0.965 | 0.930 | 0.955 | 0.955 | 0.915 | 0.940 | 0.945 |
| 60 | 0.930 | 0.955 | 0.965 | 0.940 | 0.955 | 0.965 | 0.930 | 0.945 | 0.950 |
| 65 | 0.930 | 0.940 | 0.965 | 0.930 | 0.945 | 0.970 | 0.905 | 0.905 | 0.925 |
| 70 | 0.915 | 0.955 | 0.955 | 0.945 | 0.960 | 0.975 | 0.910 | 0.925 | 0.970 |
| 75 | 0.920 | 0.955 | 0.965 | 0.940 | 0.970 | 0.965 | 0.940 | 0.910 | 0.935 |
| 80 | 0.930 | 0.960 | 0.960 | 0.925 | 0.960 | 0.970 | 0.910 | 0.890 | 0.925 |
| 85 | 0.925 | 0.950 | 0.955 | 0.930 | 0.960 | 0.960 | 0.925 | 0.940 | 0.925 |
| 90 | 0.925 | 0.955 | 0.955 | 0.940 | 0.970 | 0.955 | 0.925 | 0.945 | 0.920 |
| 95 | 0.935 | 0.960 | 0.965 | 0.930 | 0.975 | 0.955 | 0.905 | 0.925 | 0.940 |
| 100 | 0.940 | 0.960 | 0.960 | 0.940 | 0.960 | 0.960 | 1.000 | 1.000 | 1.000 |

Table 19: Performance results of model difference in mean uplift estimator results for scenario 0 at different population sizes and selection percentiles, with treatment ratio equal to 75%.

| population size percentile | model diff bias (SE) | | |
|---|---|---|---|
| | 200000 | 400000 | 800000 |
| 5 | -0.013(0.092) | -0.007(0.057) | -0.002(0.044) |
| 10 | -0.001(0.063) | -0.004(0.041) | 0.001(0.032) |
| 15 | 0.000(0.049) | -0.002(0.035) | 0.002(0.025) |
| 20 | 0.000(0.042) | -0.003(0.031) | 0.002(0.022) |
| 25 | 0.001(0.038) | -0.002(0.026) | 0.001(0.018) |
| 30 | 0.000(0.034) | -0.003(0.022) | 0.001(0.015) |
| 35 | 0.001(0.030) | -0.002(0.019) | 0.001(0.014) |
| 40 | 0.000(0.024) | -0.002(0.015) | 0.001(0.012) |
| 45 | 0.000(0.021) | -0.002(0.013) | 0.001(0.009) |
| 50 | 0.001(0.017) | -0.001(0.012) | 0.001(0.008) |
| 55 | 0.000(0.015) | -0.000(0.011) | 0.001(0.007) |
| 60 | 0.000(0.013) | -0.000(0.009) | 0.001(0.006) |
| 65 | -0.000(0.011) | -0.000(0.008) | 0.001(0.005) |
| 70 | -0.000(0.010) | 0.000(0.007) | 0.001(0.004) |
| 75 | -0.000(0.007) | -0.000(0.006) | 0.000(0.004) |
| 80 | -0.001(0.006) | -0.000(0.005) | 0.000(0.003) |
| 85 | -0.000(0.005) | -0.000(0.004) | 0.000(0.003) |
| 90 | -0.000(0.004) | -0.000(0.003) | 0.000(0.002) |
| 95 | -0.000(0.003) | -0.000(0.002) | 0.000(0.001) |
| 100 | 0.000(0.000) | 0.000(0.000) | 0.000(0.000) |

## A.3 ADDITIONAL DETAILS FOR REAL DATA APPLICATIONS

In addition to the Criteo data application, we also apply the proposed method to the MegaFon Uplift Competition Dataset provided by MegaFon at the MegaFon Uplift Competition hosted in 2021. This is a generated synthetic dataset with 50 anonymized features, a binary treatment vari-

Table 20: Coverage probability results for scenario 1 at different population sizes with treatment ratio equal to 75%.

| population size percentile | model 1 | | | model 2 | | | model diff | | |
|---|---|---|---|---|---|---|---|---|---|
| | 200000 | 400000 | 800000 | 200000 | 400000 | 800000 | 200000 | 400000 | 800000 |
| 5 | 0.950 | 0.945 | 0.945 | 0.975 | 0.895 | 0.930 | 0.970 | 0.900 | 0.940 |
| 10 | 0.960 | 0.980 | 0.975 | 0.895 | 0.940 | 0.950 | 0.925 | 0.935 | 0.950 |
| 15 | 0.955 | 0.960 | 0.935 | 0.930 | 0.940 | 0.945 | 0.930 | 0.950 | 0.935 |
| 20 | 0.950 | 0.920 | 0.930 | 0.940 | 0.925 | 0.955 | 0.925 | 0.945 | 0.935 |
| 25 | 0.945 | 0.930 | 0.960 | 0.920 | 0.945 | 0.955 | 0.940 | 0.940 | 0.950 |
| 30 | 0.935 | 0.945 | 0.955 | 0.905 | 0.930 | 0.950 | 0.930 | 0.955 | 0.915 |
| 35 | 0.940 | 0.940 | 0.955 | 0.905 | 0.940 | 0.950 | 0.930 | 0.935 | 0.920 |
| 40 | 0.945 | 0.935 | 0.955 | 0.935 | 0.915 | 0.955 | 0.930 | 0.930 | 0.920 |
| 45 | 0.930 | 0.935 | 0.965 | 0.940 | 0.910 | 0.960 | 0.950 | 0.920 | 0.915 |
| 50 | 0.925 | 0.940 | 0.970 | 0.940 | 0.915 | 0.960 | 0.955 | 0.920 | 0.910 |
| 55 | 0.945 | 0.920 | 0.960 | 0.945 | 0.925 | 0.950 | 0.975 | 0.960 | 0.925 |
| 60 | 0.940 | 0.920 | 0.955 | 0.930 | 0.935 | 0.945 | 0.955 | 0.945 | 0.930 |
| 65 | 0.920 | 0.940 | 0.950 | 0.920 | 0.915 | 0.960 | 0.940 | 0.925 | 0.955 |
| 70 | 0.930 | 0.900 | 0.975 | 0.910 | 0.900 | 0.965 | 0.915 | 0.910 | 0.955 |
| 75 | 0.930 | 0.905 | 0.965 | 0.915 | 0.900 | 0.965 | 0.920 | 0.920 | 0.950 |
| 80 | 0.920 | 0.905 | 0.955 | 0.905 | 0.900 | 0.950 | 0.920 | 0.935 | 0.935 |
| 85 | 0.940 | 0.905 | 0.965 | 0.895 | 0.925 | 0.965 | 0.925 | 0.965 | 0.930 |
| 90 | 0.915 | 0.915 | 0.960 | 0.940 | 0.920 | 0.965 | 0.915 | 0.955 | 0.965 |
| 95 | 0.915 | 0.915 | 0.950 | 0.915 | 0.920 | 0.955 | 0.905 | 0.940 | 0.930 |
| 100 | 0.915 | 0.915 | 0.965 | 0.915 | 0.915 | 0.965 | 1.000 | 1.000 | 1.000 |

Table 21: Performance results of model difference in mean uplift estimator results for scenario 1 at different population sizes and selection percentiles, with treatment ratio equal to 75%.

| population size percentile | model diff bias (SE) | | |
|---|---|---|---|
| | 200000 | 400000 | 800000 |
| 5 | 0.003(0.036) | -0.003(0.028) | -0.001(0.019) |
| 10 | 0.002(0.026) | -0.002(0.018) | -0.000(0.012) |
| 15 | 0.002(0.021) | -0.001(0.014) | 0.000(0.010) |
| 20 | 0.002(0.017) | -0.002(0.012) | -0.000(0.009) |
| 25 | 0.002(0.015) | -0.002(0.010) | -0.000(0.007) |
| 30 | 0.002(0.014) | -0.001(0.009) | 0.000(0.007) |
| 35 | 0.002(0.012) | -0.001(0.008) | 0.000(0.006) |
| 40 | 0.001(0.010) | -0.001(0.007) | 0.000(0.005) |
| 45 | 0.001(0.008) | -0.001(0.006) | 0.000(0.004) |
| 50 | 0.001(0.007) | -0.001(0.005) | 0.000(0.004) |
| 55 | 0.000(0.006) | -0.000(0.004) | 0.000(0.003) |
| 60 | -0.000(0.005) | -0.000(0.004) | 0.000(0.003) |
| 65 | -0.001(0.005) | -0.000(0.003) | 0.000(0.002) |
| 70 | -0.000(0.004) | -0.000(0.003) | 0.000(0.002) |
| 75 | -0.000(0.004) | -0.000(0.002) | -0.000(0.002) |
| 80 | -0.000(0.003) | -0.000(0.002) | -0.000(0.001) |
| 85 | -0.000(0.002) | -0.000(0.001) | 0.000(0.001) |
| 90 | -0.000(0.002) | 0.000(0.001) | -0.000(0.001) |
| 95 | -0.000(0.001) | -0.000(0.001) | 0.000(0.001) |
| 100 | 0.000(0.000) | 0.000(0.000) | 0.000(0.000) |

able and a binary outcome variable representing customer purchase. There are in total 600,000 rows in the dataset and the mean response is 0.204. The treatment assignment is random with treatment ratio equal to 0.5. The data is publicly available through the Python scikit-uplift package at `https://www.uplift-modeling.com/en/latest/api/datasets/fetch_megafon.html#megafon-uplift-competition-dataset`.

We also try to build three uplift models using XGBoost, neural networks and logistic regression base models with the same settings used in the Criteo data application. For the two-step sampling scheme, we take 60,000 (10%) of the population for simple random sample and another 60,000 for rank-based selection using model 1's ranking.

The estimated uplift curves are displayed in Figure 5. Here we can see on this dataset, there seems to be clear separations of the uplift curves among these 3 models at various selection percentiles, and the neural network model (model 2) tends to outperform the other two.

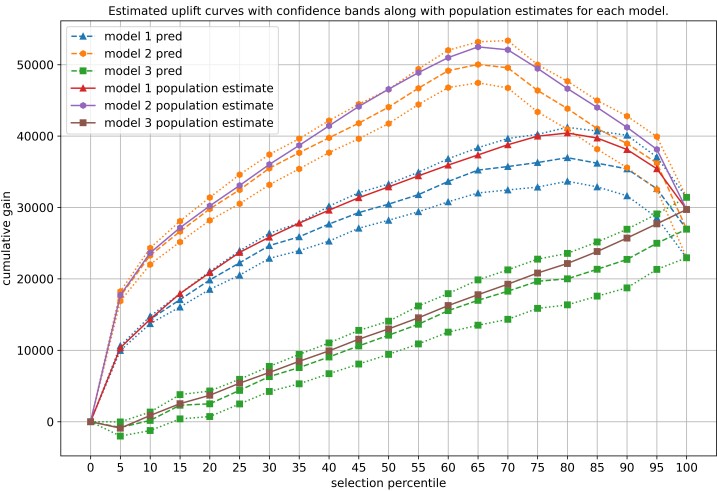

Figure 5: Uplift curve estimates for the MegaFon data.

Upon checking the estimated mean uplift differences between model 1 and 2 in Figures 6, we can see that the point-wise 95% confidence band is entirely at or below 0, which could provide strong evidence to support that model 2 is a better candidate than model 1 at any pre-specified selection percentile except the trivial 0th and 100th selection percentile cases.

The model difference comparison of model 1 vs. model 3 and model 2 vs. model 3 are included in Figures 10 and 9 in the Appendix Section A.3.1.

### A.3.1 ADDITIONAL FIGURES FOR REAL DATA APPLICATIONS

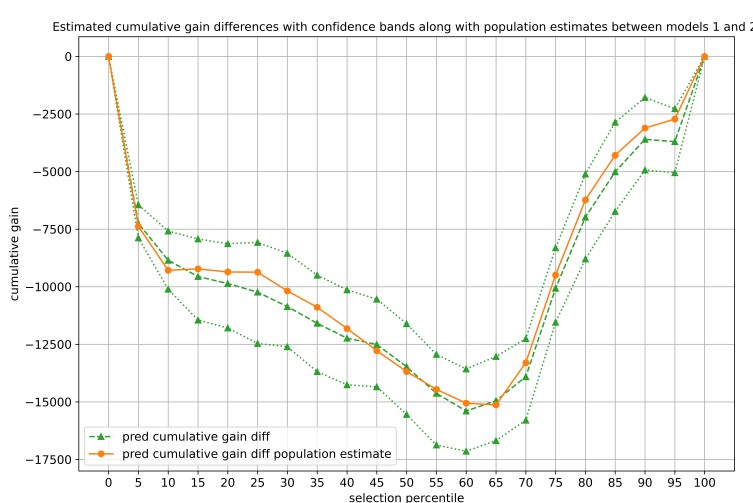

Figure 6: Estimated uplift curve differences between models 1 and 2 for the MegaFon data.

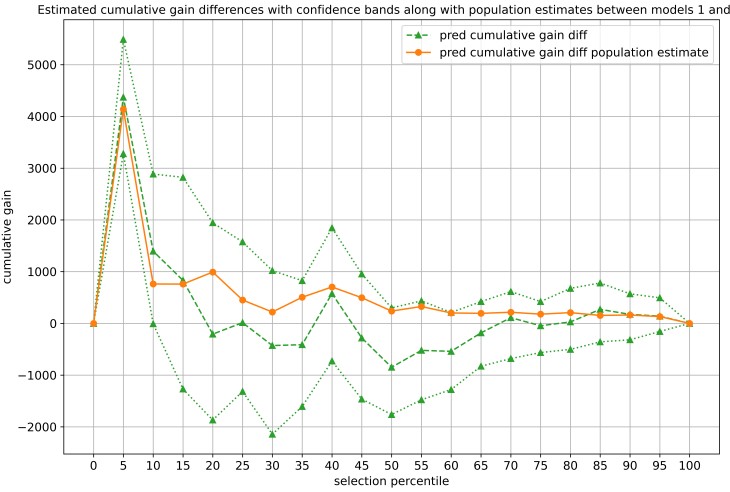

Figure 7: Estimated uplift curve differences between models 1 and 3 for the Criteo data.

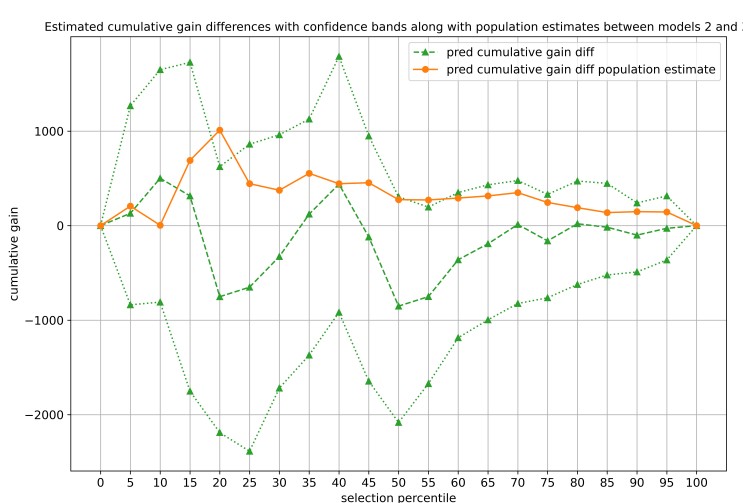

Figure 8: Estimated uplift curve differences between models 2 and 3 for the Criteo data.

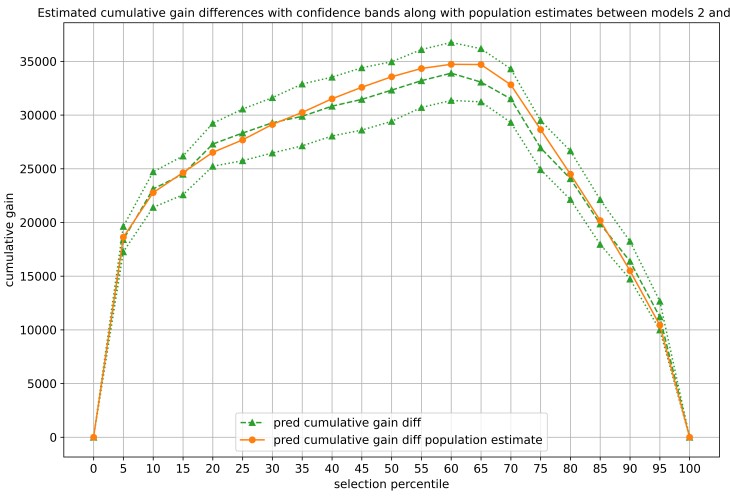

Figure 9: Estimated uplift curve differences between models 2 and 3 for the MegaFon data.

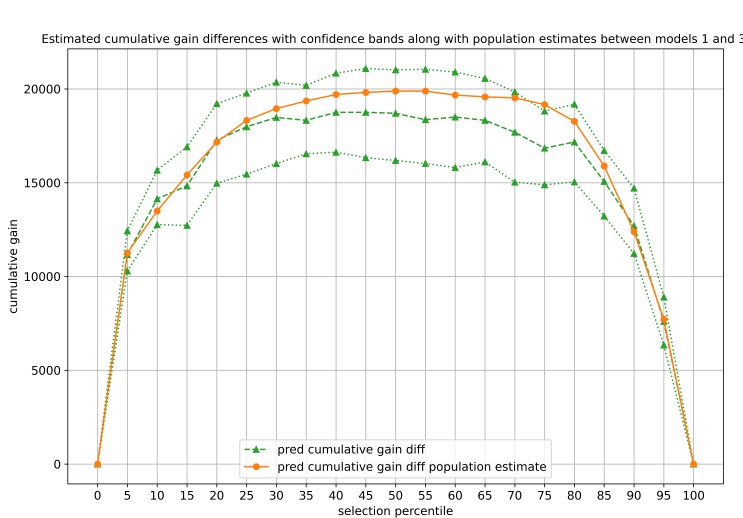

Figure 10: Estimated uplift curve differences between models 1 and 3 for the MegaFon data.

