# OpenReview forum: "We Have It Covered: A Resampling-based Method for Uplift Model Comparison"
_ICLR.cc/2026/Conference — Submitted to ICLR 2026_

### Official Review · Reviewer_3JN3 · 2025-10-15

**Soundness:** 2
**Presentation:** 1
**Contribution:** 2
**Rating:** 2
**Confidence:** 2

**Summary:**

The authors propose a resampling-based approach for quantifying the inference uncertainty in uplift curve. They claim the statistical difficulty for such uncertainty quantification, due to the impossibility of completely random sampling from population, as the x-axis of uplift curve determines how the sub-population is sampled, resulting in difference between sub- and entire populations.

Overall, I feel that the problem setup is important. However, due to the clarity issues and the lack of related work, the paper itself is very hard to follow. Moreover, the experiments lack naive baselines, such as the interval estimators under the i.i.d. assumptions, and ablation studies for comparing with e.g., naive Boostrapping. Thus, it is unclear how the proposed approach is effective in practice.

**Strengths:**

- Problem setup is important: Uncertainty quantification for uplift models is crucial for reliable decision making.
- The authors illustrate several examples to claim the statistical difficulty for this problem.

**Weaknesses:**

(A) No baseline

Due to the lack of baselines, it is impossible how greatly the proposed sampling approach is effective. Although the authors say “However, this i.i.d. assumption may not hold, which invalidates the asymptotic confidence interval obtained above.”, this invalidity is unclear, as there is no empirical comparison in the main paper.

- Is it impossible just to compare their method with such naive baselines? Coverage and length can be evaluated for such naive estimators, right?

(B) No related work section

The paper is very hard to follow at the beginning of Abstract and Introduction.

- First paragraph of Section 1: Although the authors suddenly split the two research streams (causal “impact” of interventions vs. HTE estimation), the difference seems unclear. Do they refer to Pearl’s framework vs. Rubin’s framework?

- Second paragraph of Section 1: Why these three categories? There are so many conditional average treatment effect (CATE; HTE in the authors’ wording) estimation methods. Also unclear what each category means. E.g., transform original outcome to new outcome, what does this mean? Are you referring to some representation-learning-based HTE estimation? Unclear.

In particular, I found the problem setup related to post-selection inference in statistics. I believe that some work addresses the statistical inference problem after top-K selection. Why don’t the authors cite literature on this? Is it unrelated?

(C) No Ablation Study

It seems that there is no ablation study for comparing the proposed sampling procedure with some naive one (e.g., just an ordinary Bootstrapping). Without ablation studies, readers cannot evaluate the effectiveness of proposed approach.


(D) No run time comparison

In general, resampling approaches are known to be computationally demanding. However, there is no computational complexity analysis in theory or no run time evaluation, making it hard to understand the efficiency of proposed approach.

**Questions:**

See the above questions in (A), …, (D).

---

> ### Author Response · Authors · 2025-11-21
>
> Thank you very much for your review and valuable comments. We have made improvements to the manuscript significantly to make the motivation clearer and the contents more coherent. We would like to address each of the raised questions below and truly appreciate your reconsideration of the revised manuscript.
>
> As two other reviewers mentioned the article is relatively easy to follow and understand, we would like to first provide a high level summary here to hopefully resolve the confusion.
> Motivated by real world applications where comparisons of uplift models are highly important, we aim to provide a reliable method to construct the point estimate and confidence intervals of the uplift curves on the entire targeting universe, when only a  subsample from the population is available to use.
> We also mentioned that such a subsample does not need to be a simple random sample of the population, although it needs to be a random sample. There is a trade-off between exploration and exploitation in an iterative model development process that’ll lead to an unequal-probability subsample of the population.
> We proposed a two-step sampling scheme to balance the real-world exploration and exploitation regarding candidate selection in practice, as well as the sample inclusion probability formulas under this scheme.
> We proposed and examined the nested bootstrap approach empirically for construction of uplift curve point estimates and confidence intervals on the entire universe
>
> Onto the mentioned weaknesses:
>
> For (A), we included a discussion of the so-called difference in difference method when we compare 2 models at a fixed selection size in practice, and the reviewer mentioned this could be used as a baseline. However, this “baseline estimate” here is not giving us the uplift curve estimate on the entire population and thus it is not a relevant baseline. As there is no prior work being done in this area, no other meaningful benchmarks to compare with. Therefore we used simulations to examine the properties of our method to show its validity.
>
> For (B), the reviewer firstly got confused on the literature review of the uplift methods as well as HTE methods, and these 2 research streams have a lot of commonalities for the underlying problem formulation and we provided another reference for a systematic review. It is not very relevant and not our intention to differentiate between Pearl’s or Rubin’s framework here.
>
> The reviewer then has questions on the three categories of uplift methods we discussed in the article. This is a common categorization by researchers and the reviewer may read the references we cited, especially (Gutierrez & G´erardy, 2017), as well as this article mentioned by another reviewer:
>
> Zhang, Weijia, Jiuyong Li, and Lin Liu. "A unified survey of treatment effect heterogeneity modelling and uplift modelling." ACM Computing Surveys (CSUR) 54.8 (2021): 1-36.
>
> The reviewer had a concern about the work being relevant to post-selection inference and not citing any top-K selection related literature. We are not aware of any highly relevant literature for the problem we are trying to address and would really appreciate it if the reviewer could point us to some references.
>
>
> For (C), we agree with the reviewer that we could potentially add results of some naive approaches here such as ordinary bootstrap. We actually tried different variations of bootstrap and the coverage probabilities were way below the nominal levels (e.g. <20% for a 95% interval) which made those methods invalid, and the performance is too inferior to form an interesting comparison.
>
>
> For (D), this is a great point which is also raised by another reviewer. On the computation complexity front, the cost of computing the uplift curve estimate using the population-level data for one model is O(Nlog(N)) due to sorting, and we conduct B outer bootstraps and D inner bootstraps for total S models, the computation complexity would be O(log(N) x N x B x D x S). Empirically, it took us less than 1 hour to conduct the analysis of the Criteo Data application on a single Amazon AWS r5.24xlarge EC2 instance with no code optimizations. We have added that into the revision.
>
>
> Again, we are very grateful for your review and insightful comments, and we would truly appreciate your reconsideration on this manuscript.

---

> > ### Comment · Reviewer_3JN3 · 2025-11-26
> >
> > I thank the authors for their detailed response. However, I will keep my rating of "2: rejection", as I do not feel my concerns are addressed, mainly due to (A) the lack of baselines and (C) comparison with naive Bootstrapping.
> >
> > Without sufficient empirical comparison, we cannot evaluate the significance of the authors' proposed framework. Verifying their added results (which I could not find in the paper PDF) will require another review to be accepted to a top conference.

---

> > > ### Author Response · Authors · 2025-11-26
> > >
> > > Thank you very much for your follow up response, and we would like to clarify more on your remaining concerns:
> > > 1) Regarding the comparison with naive Bootstrapping: The original naive bootstrap method requires a simple random sample from the population, i.e., all the data points in the sample need to be independently and identically distributed (i.i.d.). This condition is not satisfied here as the random sample we have could be drawn with unequal probability, due to the exploration and exploitation trade-off in practice. The naive bootstrap have significantly inferior simulation results during our exploration to render the comparison uninformative, as in this case it is not even theoretically valid in this situation.
> > >
> > > 2) Regarding lack of baselines: As we mentioned in the introduction, there is no existing literature available that's highly relevant to this topic, and our main contribution is to provide an approach and show its validity (through coverage probability simulations). It could be used as a baseline approach for future research in this field, and we firmly believe this contribution is significant.
> > >
> > > We truly appreciate your reconsideration of the manuscript again, and feel free to let us know if you have further questions. Thank you very much for your time and help!

---

> > > > ### Author Response · Authors · 2025-11-29
> > > >
> > > > As indicated in the latest ICLR email that there will be no further discussion from the reviewers, we have added more clarifications to the latest revision to address all the raised comments and weaknesses. We sincerely appreciate the reviewers’ thorough evaluations and thoughtful feedback, and we hope the area chair will consider our manuscript based on its merits and contributions to the research area.

---

### Official Review · Reviewer_7qu8 · 2025-10-23

**Soundness:** 3
**Presentation:** 2
**Contribution:** 2
**Rating:** 4
**Confidence:** 3

**Summary:**

The authors propose a two-step sampling design and nested bootstrap procedure to estimate the uplift curve for the whole population with UQ using a population subsample. The method first draws a random subset of the population and then adds a sample of top-ranked individuals to ensure coverage of high-uplift segments. Using inverse-probability weighting, they estimate the theoretical uplift curve from this subsample, and apply a nested bootstrap for uncertainty quantification.

**Strengths:**

- the paper addresses a relevant and challenging topic of valuating uplift models on the full population using only a subsample
- the two-step sampling and weighting idea is simple and intuitive, easy to implement
- the nested bootstrap provides a practical way to compute confidence intervals for uplift curves
- the authors test the method on both synthetic and real-world data

**Weaknesses:**

- I think the paper writing is not very easy to follow and could be organized more clearly in my opinion. For instance, I would have appreciated a more intuitive introduction of the nested bootstrap
- the authors assume that the treatment assignment $T$ is independent of $X$, which is plausible in randomized experiments. This should be better emphasized as limits the applicability of the setting.
- I am not sure whether this is a sufficient contribution. I still have some doubts, see below.

Minor:
- Plots 3,4 are hard not visually nice and hard to read. The text should be bigger and I would recommend putting more time into the layout improvement

**Questions:**

- this nested bootstrap procedure is very computationally heavy, can you elaborate on the computational cost of the algorithm? I think this should be added to the paper
- empirically, how robust is the method if treatment assignment is not fully randomized, for instance if there is some form of confounding?
- can the authors elaborate on how the variance of the estimator scales with the inclusion probabilities and sample size, and whether any finite-sample hold?

---

> ### Author Response · Authors · 2025-11-21
>
> Thank you very much for your review and insightful comments. We have revised the manuscript based on your suggestions, and we will address the weaknesses and questions mentioned in the review below.
>
> In the situation of not fully randomized treatment assignment, the proposed approach likely needs some changes to make sure we have valid uplift estimates, and needs future work. We limit the scope of the manuscript to randomized treatment allocation, as it is common in marketing applications such as direct mail marketing and online product A/B testing when the marketer could determine the randomization. The proposed method could be useful and relevant for real world applications.
>
> As the uplift curve’s nested bootstrap point estimator involves inclusion probabilities of all the samples with the constraint to sum up to 1, we haven’t thought about how the variance changes with different inclusion probability distributions among samples. Empirically, we have included a discussion in Appendix section A.2.1, where we found that when the total selection size is the same, the estimator constructed with a bigger simple random sample has lower standard error. In addition, the standard error seems to scale with a rate of 1 over square root of the sample size roughly.
>
> When sample size is very limited, our simulation results in the appendix show subpar coverage results, however this is a known limitation to bootstrap.
>
> On the computation complexity front, the cost of computing the uplift curve estimate using the population-level data for one model is O(Nlog(N)) due to sorting, and we conduct B outer bootstraps and D inner bootstraps for total S models, the computation complexity would be O(log(N) x N x B x D x S). Empirically, it took us less than 1 hour to conduct the analysis of the Criteo Data application on a single Amazon AWS r5.24xlarge EC2 instance with no code optimizations. We have added that into the revision.
>
> We have also revised the section of the nested bootstrap in the revision to give more motivation as well as detailed intuition.
>
> Thank you very much again for your review and valuable feedbacks, and we truly appreciate your reconsideration of the revised manuscript.

---

> > ### Author Response · Authors · 2025-11-29
> >
> > As indicated in the latest ICLR email that there will be no further discussion from the reviewers, we have added more clarifications to the latest revision to address all the raised comments and weaknesses. We sincerely appreciate the reviewers’ thorough evaluations and thoughtful feedback, and we hope the area chair will consider our manuscript based on its merits and contributions to the research area.

---

### Official Review · Reviewer_GdSY · 2025-10-28

**Soundness:** 1
**Presentation:** 3
**Contribution:** 2
**Rating:** 2
**Confidence:** 3

**Summary:**

In this paper, the authors propose a method to construct uplift curve estimates for the entire population using a population subsample. This approach primarily relies on combining data randomly sampled from the entire population with the subsamples and utilizing Inverse Probability Weighting (IPW) to eliminate selection bias between the subsamples and the entire population. The authors conduct experiments on both synthetic and real-world datasets to validate the effectiveness of the proposed method.

**Strengths:**

1. The paper is well-organized and easy to follow.
2. The authors provide a diverse set of scenarios in the simulation study, and the extensive experimental results show the robustness of the proposed method.
3. The use of real-world datasets adds significant credibility and practical relevance to the proposed method.
4. The experimental details are thoroughly presented, which should make the results easy to replicate.

**Weaknesses:**

1. The authors' understanding of the uplift model evaluation in industrial applications seems to deviate from practical realities. The authors claim that the dataset used for uplift model evaluation (the "subsample of the entire population") needs to be firstly sorted by the model scores and then top-x% samples selected for treatment-control division. Such sample selection process introduces selection bias between the subsample and the entire population, and thus they propose a method to address this issue. However, in practical industrial applications, it is more common to directly apply random treatment-control assignment to the entire subsample through A/B tests, rather than using the top-x% “sub-subsamples”. The uplift curve is then evaluated over the entire subsample by ranking based on the model scores [1]. Therefore, it is questionable whether the selection bias between the subsample and the entire population, as the primary challenge to address in the paper, is a common issue in real-world applications.

2. The authors overlook a significant body of work that addresses the selection bias between subsamples and the entire population [2]. The approach the authors use is closely related to the commonly applied inverse probability weighting (IPW) methods in this area, where they explicitly define the inclusion probability rather than estimating it via a model, as is typically done in related works. While the idea is good, the lack of citations to relevant literature is a concern and should be addressed.

3. The experiments only consider the scenarios with only three models to be evaluated, but in real-world applications, there are often many more models as candidates. It remains unclear whether the algorithm's performance and efficiency would hold up under a larger set of models. The authors should consider adding experiments with more models to validate the scalability and robustness of their method.

[1] Zhang, Weijia, Jiuyong Li, and Lin Liu. "A unified survey of treatment effect heterogeneity modelling and uplift modelling." ACM Computing Surveys (CSUR) 54.8 (2021): 1-36.

[2] Colnet, Bénédicte, et al. "Causal inference methods for combining randomized trials and observational studies: a review." Statistical science 39.1 (2024): 165-191.

**Questions:**

Please see Weaknesses. My primary concern lies in Weaknesses 1 and 2. I would appreciate it if the authors could provide further clarification on the specific scenarios where the problem studied in this paper would apply.

---

> ### Author Response · Authors · 2025-11-21
>
> We are very grateful for your thoughtful review and questions on the manuscript. We will address the Weaknesses and questions below, and we truly appreciate your reconsideration of the revised manuscript.
>
> On Weakness 1, your main concern seems to be whether in real-world applications it is common to do a rank-based selection with uplift models first, and then only take a subsample from the population, compared to assigning treatment-control randomly on the entire population or a simple random sample of the population. First, for any industrial applications where marketing interventions have some unit cost structure (such as direct mail marketing or advertisements charged per user) and a fixed budget, it is infeasible to test the treatment/control options on the entire universe. If only a small percentage of the entire universe are profitable candidates, it could be quite costly and difficult in practice to take a large enough simple random sample for reliable uplift curve constructions.
>
> In addition, there is a trade-off between exploration and exploitation in an iterative model development process. One may afford to collect one simple random sample to develop models and identify some promising candidate uplift models. Then in subsequent future marketing campaigns, model-rank based selections from the top candidate model(s) are often used to optimize for uplift/incremental profits. For model comparison in production, it is common to split the selection universe randomly, and use different candidate models to rank and select the top candidates, then try to compare the real-world uplift performance of these models at given selection size. This will go back to the problem described in the manuscript, that we won’t be able to recover the uplift curve on the entire universe for multiple models, and perform uncertainty quantification without selecting a random sample (not necessarily a simple random sample) from the population.
>
> We appreciate your comments on this and we have revised the article to articulate the practical considerations and our goals more clearly.
>
> On Weakness 2, first of all we want to thank the reviewer for sharing the reference on IPW and we have added it into the manuscript. Meanwhile, we would like to clarify that our nested bootstrap approach may have some spirit of the IPW approach, it is intrinsically different from the IPW approach in terms of how the point estimate as well as the confidence interval are constructed. If one tries to construct the IPW point estimate for the uplift curve, the form could be rather complicated and a theoretical derivation of the confidence interval is even more difficult if possible. To the best of our knowledge, there is no existing literature discussing the feasibility and validity of using IPW for uplift curve estimation and uncertainty quantification.
>
> As we mentioned earlier on the exploit phase of real world marketing applications, the candidate selection process tends to be mainly driven by uplift/model ranks, rather than determined by random choice with pre-specified inclusion probabilities described in the referred literature. It is not trivial to derive the inclusion probability formula for selection schemes involving model-ranks and one may choose to estimate it with a model if closed form is not available.
>
> It’s worth noting that in our manuscript, we derived closed-form formulas for the inclusion probabilities with 1 or more models for model-based selections under the proposed two-step sample selection scheme, instead of using a model to estimate them. The proposed sampling scheme could be highly useful for production model comparisons and iterative model development as it provides a way to balance exploration and exploitation in practice.
>
>
> On Weakness 3, we only illustrated 3 models in the real data applications to showcase the proposed method, although our derivation and implementation is not limited to just 3 models. Nevertheless, when there are many candidate models we want to compare in production, one could expect many sets of pair-wise model comparisons and the need for multiplicity adjustment similar to statistical tests with multiple variants. This is a separate topic from the focus of this article.
>
> Thank you very much again for your constructive feedbacks, and we really appreciate your reconsideration of the revised manuscript.

---

> > ### Author Response · Authors · 2025-11-29
> >
> > As indicated in the latest ICLR email that there will be no further discussion from the reviewers, we have added more clarifications to the latest revision to address all the raised comments and weaknesses. We sincerely appreciate the reviewers’ thorough evaluations and thoughtful feedback, and we hope the area chair will consider our manuscript based on its merits and contributions to the research area.

---

### Official Review · Reviewer_DTAk · 2025-10-30

**Soundness:** 3
**Presentation:** 2
**Contribution:** 3
**Rating:** 6
**Confidence:** 3

**Summary:**

This paper studies how to address the gap of estimating full-population uplift curves using only subsamples.A two-step sampling scheme and a nested bootstrap approach are proposed. It is validated by both simulations and real datasets (Criteo, MegaFon), where the proposed method successfully identifies meaningful model differences, which provides a reliable tool for uplift model comparison.

**Strengths:**

1. It is the first to propose a framework for estimating full-population uplift curves using subsamples, with explicit uncertainty quantification.
2. Rigorous simulation and real-data experiments confirm the method’s validity (coverage, low bias) and ability to identify meaningful model differences.

**Weaknesses:**

The major weakness is the presentation, especially the problem formulation part. The contents seem to be a combination of several issues without clear connection, making it hard to follow.

**Questions:**

The Qini coefficient and uplift curve are only suitable for the cases of single action (T \in {0,1}) and binary response (Y \in {0,1}), is it possible to extend the proposed approach for more general settings, e.g. multiple actions and response in real numbers?

---

> ### Author Response · Authors · 2025-11-21
>
> Thank you very much for your thoughtful review and comments. We have revised the article presentation to emphasize our main methodology contribution and make the contents more coherent. We would really appreciate your reconsideration of the presentation aspect of the revised manuscript.
>
> Regarding your question on the possible extensions, our proposed methodology can be naturally extended to numerical response variables as the procedure doesn’t require Y to be binary. For multiple actions, we haven’t examined it yet as it is a separate a more difficult problem, and it could be a very interesting future direction.

---

> > ### Author Response · Authors · 2025-11-29
> >
> > As indicated in the latest ICLR email that there will be no further discussion from the reviewers, we have added more clarifications to the latest revision to address all the raised comments and weaknesses. We sincerely appreciate the reviewers’ thorough evaluations and thoughtful feedback, and we hope the area chair will consider our manuscript based on its merits and contributions to the research area.

---

### Meta-Review · Area_Chair_VWH9 · 2026-01-04

**Summary:**

While the paper raises a relevant practical issue in uplift modeling under subsampling constraints, the methodological contribution is limited and largely incremental, with key ideas (two-step sampling and resampling-based uncertainty) closely related to existing survey sampling and off-policy evaluation techniques. The theoretical justification is underdeveloped, and the empirical evaluation does not convincingly demonstrate advantages over simpler baselines or clarify when the proposed approach is necessary or optimal.

**Reviewer Concerns:**

Nothing to note

**Reviewer Scores:**

I can't predict this.

---

### Decision · Program_Chairs · 2026-01-26

Reject